# No Training, No Problem: Rethinking Classifier-Free Guidance for Diffusion Models

**Seyedmorteza Sadat[1], Manuel Kansy[1], Otmar Hilliges[1], Romann M. Weber[2]**
[1]ETH Zürich, [2]DisneyResearch|Studios
`{seyedmorteza.sadat, manuel.kansy, otmar.hilliges}@inf.ethz.ch`
`{romann.weber}@disneyresearch.com`

## Abstract

Classifier-free guidance (CFG) has become the standard method for enhancing the quality of conditional diffusion models. However, employing CFG requires either training an unconditional model alongside the main diffusion model or modifying the training procedure by periodically inserting a null condition. There is also no clear extension of CFG to unconditional models. In this paper, we revisit the core principles of CFG and introduce a new method, independent condition guidance (ICG), which provides the benefits of CFG without the need for any special training procedures. Our approach streamlines the training process of conditional diffusion models and can also be applied during inference on any pre-trained conditional model. Additionally, by leveraging the time-step information encoded in all diffusion networks, we propose an extension of CFG, called time-step guidance (TSG), which can be applied to *any* diffusion model, including unconditional ones. Our guidance techniques are easy to implement and have the same sampling cost as CFG. Through extensive experiments, we demonstrate that ICG matches the performance of standard CFG across various conditional diffusion models. Moreover, we show that TSG improves generation quality in a manner similar to CFG, without relying on any conditional information.

## 1 Introduction

Diffusion models have recently emerged as the main methodology behind many successful generative models (Sohl-Dickstein et al., 2015; Ho et al., 2020; Dhariwal & Nichol, 2021; Rombach et al., 2022; Song & Ermon, 2019; Song et al., 2021b). At the core of such models lies a diffusion process that gradually adds noise to the data until the corrupted points are indistinguishable from pure noise. During inference, a denoiser trained to reverse this process is used to gradually refine pure-noise samples until they resemble the clean data. While the theory suggests that standard sampling from diffusion models should yield high-quality images, this does not generally hold in practice, and guidance methods are often required to increase the quality of generations, albeit at the expense of diversity (Dhariwal & Nichol, 2021; Ho & Salimans, 2022). Classifier guidance (Dhariwal & Nichol, 2021) introduced this quality-boosting concept by utilizing the gradient of a classifier trained on noisy images to increase the class-likelihood of generated samples. Later, classifier-free guidance (CFG) (Ho & Salimans, 2022) was proposed, allowing diffusion models to simulate the same behavior as classifier guidance without using an explicit classifier. Since then, CFG has been applied to other conditional generation tasks, such as text-to-image synthesis (Nichol et al., 2022) and text-to-3D generation (Poole et al., 2023).

In addition to CFG's trading diversity for quality, it has the following two practical limitations. First, it requires a dedicated, pre-defined training process on an *auxiliary* task in order to learn the unconditional score function. This typically involves training a separate unconditional model or, more commonly, randomly dropping the conditioning vector and replacing it with a null vector during training. This approach reduces training efficiency, as the model now needs to be trained on two different tasks. Moreover, replacing the condition may not be straightforward when multiple conditioning signals—such as text, images, and audio—are used simultaneously or when the null vector (often the zero vector) carries specific meaning. We demonstrate that this dedicated auxiliary training process is unnecessary. A second limitation is that there has been no clear way to extend

the benefits of classifier-free guidance beyond conditional models to unconditional generation. We introduce a method that closes this gap.

We revisit the principles behind classifier-free guidance and show both theoretically and empirically that similar quality-boosting behavior can be achieved without the need for additional auxiliary training of an unconditional model. The main idea is that by using a conditioning vector *independent* of the input data, the conditional score function becomes equivalent to the unconditional score. This insight leads us to propose *independent condition guidance* (ICG), a method that replicates the behavior of CFG at inference time without requiring auxiliary training of an unconditional model, i.e., without needing explicit access to the unconditional score function. In Section 6.1, we show that the auxiliary training of the unconditional model in CFG can be detrimental to training efficiency, and similar or better performance can be achieved by training only a purely conditional model and using ICG instead.

Inspired by the above, we also introduce a novel technique to extend classifier-free guidance to a more general setting that includes unconditional generation. We argue that by using a perturbed version of the time-step embedding in diffusion models, one can create a guidance signal similar to CFG to improve the quality of generations. This method, which we call *time-step guidance* (TSG), aims to improve the accuracy of denoising at each sampling step by leveraging the time-step information learned by the diffusion model to steer sampling trajectories toward better noise-removal paths.

ICG and TSG are easy to implement, do not require additional fine-tuning of the underlying diffusion models, and have the same sampling cost as CFG. Through extensive experiments, we empirically verify that: 1) ICG offers performance similar to CFG and can be readily applied to models that are not trained with the CFG objective in mind, such as EDM (Karras et al., 2022); and 2) TSG improves output quality in a manner similar to CFG for both conditional and unconditional generation.

The core contributions of our work are as follows: (i) We revisit the principles of classifier-free guidance and offer an efficient, theoretically motivated method to employ CFG without requiring any auxiliary training of an unconditional model, greatly simplifying the training process of conditional diffusion models and improving training efficiency relative to the standard approach. (ii) We offer an extension of CFG that is generally applicable to all diffusion models, whether conditional or unconditional. (iii) We demonstrate empirically that our guidance techniques achieve the quality-boosting benefits of CFG across various setups and network architectures.

## 2 RELATED WORK

Score-based diffusion models (Song & Ermon, 2019; Song et al., 2021b; Sohl-Dickstein et al., 2015; Ho et al., 2020) learn the data distribution by reversing a forward diffusion process that progressively transforms the data into Gaussian noise. These models have quickly surpassed the fidelity and diversity of previous generative modeling methods (Nichol & Dhariwal, 2021; Dhariwal & Nichol, 2021), achieving state-of-the-art results in various domains, including unconditional image generation (Dhariwal & Nichol, 2021; Karras et al., 2022), text-to-image generation (Ramesh et al., 2022; Saharia et al., 2022b; Balaji et al., 2022; Rombach et al., 2022; Podell et al., 2023; Yu et al., 2022), video generation (Blattmann et al., 2023b;a; Gupta et al., 2023), image-to-image translation (Saharia et al., 2022a; Liu et al., 2023), motion synthesis (Tevet et al., 2023; Tseng et al., 2023), and audio generation (Chen et al., 2021; Kong et al., 2021; Huang et al., 2023).

Since the development of the DDPM model (Ho et al., 2020), many advancements have been proposed including improved network architectures (Hoogeboom et al., 2023; Karras et al., 2023; Peebles & Xie, 2022; Dhariwal & Nichol, 2021), sampling algorithms (Song et al., 2021a; Karras et al., 2022; Liu et al., 2022; Lu et al., 2022a; Salimans & Ho, 2022), and training methods (Nichol & Dhariwal, 2021; Karras et al., 2022; Song et al., 2021b; Salimans & Ho, 2022; Rombach et al., 2022). Despite these recent advances, diffusion guidance, including classifier and classifier-free guidance (Dhariwal & Nichol, 2021; Ho & Salimans, 2022), still plays an essential role in improving the quality of generations as well as increasing the alignment between the condition and the output image (Nichol et al., 2022).

SAG (Hong et al., 2022) and PAG (Ahn et al., 2024) have recently been proposed to increase the quality of UNet-based diffusion models by modifying the predictions of the self-attention layers. Our method is complementary to these approaches, as one can combine ICG updates with the update

signal from the perturbed attention modules (Hong et al., 2022). In addition, we make no assumptions about the network architecture.

Another line of work includes guiding the generation of the diffusion model with a differentiable loss function or an off-the-shelf classifier (Song et al., 2023; Chung et al., 2022; Yu et al., 2023; Bansal et al., 2023; He et al., 2023). These methods are primarily focused on solving inverse problems, typically with unconditional models, while we are instead concerned with achieving the benefits of CFG in conditional models without any additional training requirements. With TSG, we also generalize our approach to extend CFG-like benefits to unconditional models.

Perturbing the condition vector is employed in CADS (Sadat et al., 2024) to increase the diversity of generations. CADS differs from ICG in focusing on the *conditional* branch to improve diversity, while ICG is concerned with the *unconditional* branch to simulate CFG. Since CADS is designed to enhance the diversity of CFG, it can be used alongside ICG to improve the diversity of output at high guidance scales (see Appendix C).

## 3 BACKGROUND

This section provides an overview of diffusion models. Let $x \sim p_{\text{data}}(x)$ be a data point, $t \in [0, 1]$ be the time step, and $z_t = x + \sigma(t)\epsilon$ be the forward process of the diffusion model that adds noise to the data. Here $\sigma(t)$ is the noise schedule and determines how much information is destroyed at each time step $t$, with $\sigma(0) = 0$ and $\sigma(1) = \sigma_{\text{max}}$. Karras et al. (2022) showed that this forward process corresponds to the ordinary differential equation (ODE)

$$\mathrm{d}z_t = -\dot{\sigma}(t)\sigma(t) \nabla_{z_t} \log p_t(z_t)\mathrm{d}t \tag{1}$$

or, equivalently, a stochastic differential equation (SDE) given by

$$\mathrm{d}z_t = -\dot{\sigma}(t)\sigma(t) \nabla_{z_t} \log p_t(z_t) \,\mathrm{d}t - \beta(t)\sigma(t)^2 \nabla_{z_t} \log p_t(z_t) \,\mathrm{d}t + \sqrt{2\beta(t)}\sigma(t) \,\mathrm{d}\omega_t. \tag{2}$$

Here $\mathrm{d}\omega_t$ is the standard Wiener process, and $p_t(z_t)$ is the time-dependent distribution of noisy samples, with $p_0 = p_{\text{data}}$ and $p_1 = \mathcal{N}(\mathbf{0}, \sigma_{\text{max}}^2 \mathbf{I})$. Assuming that we have access to the time-dependent score function $\nabla_{z_t} \log p_t(z_t)$, we can sample from the data distribution $p_{\text{data}}$ by solving the ODE or SDE backward in time (from $t = 1$ to $t = 0$). The unknown score function $\nabla_{z_t} \log p_t(z_t)$ is estimated via a neural denoiser $D_\theta(z_t, t)$ that is trained to predict the clean samples $x$ from the corresponding noisy samples $z_t$. The framework allows for conditional generation by training a denoiser $D_\theta(z_t, t, y)$ that accepts additional input signals $y$, such as class labels or text prompts.

**Training objective**  Given a noisy sample $z_t$ at time step $t$, the denoiser $D_\theta(z_t, t, y)$ with parameters $\theta$ can be trained with the standard MSE loss (also called denoising score matching loss)

$$\arg\min_\theta \mathbb{E}_t\Big[\|D_\theta(z_t, t, y) - x\|^2\Big]. \tag{3}$$

The denoiser approximates the time-dependent conditional score function $\nabla_{z_t} \log p_t(z_t|y)$ via

$$\nabla_{z_t} \log p_t(z_t|y) \approx \frac{D_\theta(z_t, t, y) - z_t}{\sigma(t)^2}. \tag{4}$$

**Classifier-free guidance (CFG)**  CFG is an inference method for improving the quality of generated outputs by mixing the predictions of a conditional and an unconditional model (Ho & Salimans, 2022). Specifically, given a null condition $y_{\text{null}} = \varnothing$ corresponding to the unconditional case, CFG modifies the output of the denoiser at each sampling step according to

$$\hat{D}_\theta(z_t, t, y) = D_\theta(z_t, t, y_{\text{null}}) + w_{\text{CFG}}(D_\theta(z_t, t, y) - D_\theta(z_t, t, y_{\text{null}})), \tag{5}$$

where $w_{\text{CFG}} = 1$ corresponds to the non-guided case. The unconditional model $D_\theta(z_t, t, y_{\text{null}})$ is trained by randomly assigning the null condition $y_{\text{null}} = \varnothing$ to the input of the denoiser with probability $p$, where we normally have $p \in [0.1, 0.2]$. One can also train a separate denoiser to estimate the unconditional score in Equation (5) (Karras et al., 2023). Similar to the truncation method in GANs (Brock et al., 2019), CFG increases the quality of individual images at the expense of less diversity (Murphy, 2023).

## 4 REVISITING CLASSIFIER-FREE GUIDANCE

We now show how a conditional model can be used to simulate the behavior of classifier-free guidance, without needing any auxiliary training to learn the unconditional score function. The analysis in this section is inspired by Sadat et al. (2024).

First, note that at each time step $t$, classifier-free guidance implicitly uses the conditional score $\nabla_{\boldsymbol{z}_t} \log p_t(\boldsymbol{z}_t|\boldsymbol{y})$ and the unconditional score $\nabla_{\boldsymbol{z}_t} \log p_t(\boldsymbol{z}_t)$ to guide the sampling process. From Bayes' theorem, we can write $p_t(\boldsymbol{z}_t|\boldsymbol{y}) = \frac{p_t(\boldsymbol{y}|\boldsymbol{z}_t)p_t(\boldsymbol{z}_t)}{p_t(\boldsymbol{y})}$, which gives us

$$\nabla_{\boldsymbol{z}_t} \log p_t(\boldsymbol{z}_t|\boldsymbol{y}) = \nabla_{\boldsymbol{z}_t} \log p_t(\boldsymbol{z}_t) + \nabla_{\boldsymbol{z}_t} \log p_t(\boldsymbol{y}|\boldsymbol{z}_t). \tag{6}$$

Next, assume that we replace the condition $\boldsymbol{y}$ with a random vector $\hat{\boldsymbol{y}} \sim q(\hat{\boldsymbol{y}})$ that is independent of the input $\boldsymbol{z}_t$. In this case, the "classifier" $p_t(\hat{\boldsymbol{y}}|\boldsymbol{z}_t)$ is equal to $q(\hat{\boldsymbol{y}})$, which gives us

$$\nabla_{\boldsymbol{z}_t} \log p_t(\boldsymbol{z}_t|\hat{\boldsymbol{y}}) \approx \nabla_{\boldsymbol{z}_t} \log p_t(\boldsymbol{z}_t) + \nabla_{\boldsymbol{z}_t} \log q(\hat{\boldsymbol{y}}) = \nabla_{\boldsymbol{z}_t} \log p_t(\boldsymbol{z}_t). \tag{7}$$

This suggests that we can estimate the unconditional score purely based on the conditional model by replacing the condition $\boldsymbol{y}$ with an independent vector $\hat{\boldsymbol{y}}$. We argue as a result that there is no need to train a separate model $D_{\boldsymbol{\theta}}(\boldsymbol{z}_t, t, \boldsymbol{y}_{\text{null}})$ to apply classifier-free guidance, as we can use the conditional model itself to predict the score of the unconditional distribution as long as we pick an input condition that is independent of $\boldsymbol{z}_t$. We call this method *independent condition guidance* (ICG) for the rest of the paper.

In reality, the scores in Equation (7) are approximated by finite-capacity parametric models on *dependent* data drawn from $p_t(\boldsymbol{z}_t, \boldsymbol{y})$.[1] In this case, $D_{\boldsymbol{\theta}}(\boldsymbol{z}_t, t, \hat{\boldsymbol{y}})$ will not necessarily exactly equal $D_{\boldsymbol{\theta}}(\boldsymbol{z}_t, t, \boldsymbol{y}_{\text{null}})$ since independent $\hat{\boldsymbol{y}}$ and $\boldsymbol{z}_t$ are technically out of distribution relative to what the model was trained on. We have not found this to be an issue in practice, and further, the expected error in the unconditional score estimate can be made arbitrarily small through both the capacity of $D_{\boldsymbol{\theta}}$ and the choice of $q(\hat{\boldsymbol{y}})$. We provide an analysis in the appendix.

**Implementation details** In practice, we experiment with two options for the independent condition. First, $\hat{\boldsymbol{y}}$ can be drawn from a Gaussian distribution with a suitable standard deviation so that $\hat{\boldsymbol{y}}$ matches the scale of the actual conditioning vector $\boldsymbol{y}$. Second, a random condition from the conditioning space, such as a random class label or random clip tokens, can be chosen as the independent $\hat{\boldsymbol{y}}$. We show in Section 7 that both methods perform similarly. However, there may be a slight preference for the random condition over Gaussian noise, as it stays closer to the conditioning distribution that the diffusion model was trained on.

## 5 TIME-STEP GUIDANCE

Inspired by ICG, we next offer an extension of classifier-free guidance that can be used with any model, including unconditional networks. We begin our analysis with class-conditional models and subsequently extend it to a more general setting.

In the class-conditional case, the embedding vector of the class is typically added to the embedding vector of the time step $t$ to compute the input condition of the diffusion network. Hence, in practice, CFG essentially uses the outputs of the diffusion network for two different input embeddings and takes their difference as the update direction. Thus, we might directly utilize the time-step embedding of each diffusion model as a means to define a similar guidance signal. This leads to a novel method that we refer to as *time-step guidance* (TSG), which, like CFG, increases the quality of generations but, unlike CFG, is applicable even to unconditional models.

In this method, we compute the model outputs for the clean time-step embedding and a perturbed embedding and use their difference to guide the sampling. More specifically, at each time step $t$, we update the output via

$$\hat{D}_{\boldsymbol{\theta}}(\boldsymbol{z}_t, t) = D_{\boldsymbol{\theta}}(\boldsymbol{z}_t, \tilde{t}) + w_{\text{TSG}}\big(D_{\boldsymbol{\theta}}(\boldsymbol{z}_t, t) - D_{\boldsymbol{\theta}}(\boldsymbol{z}_t, \tilde{t})\big), \tag{8}$$

where $\tilde{t}$ is the perturbed version of $t$. The intuition behind TSG is that at each time step $t$, altering the time-step embedding of the network leads to denoised outputs with either insufficient or excessive

---

[1]However, at the highest noise scales, $p_t(\boldsymbol{z}_t, \boldsymbol{y}) \approx p_t(\boldsymbol{z}_t)p(\boldsymbol{y})$.

noise removal (see Figure 10 in the appendix). Consequently, these outputs can be exploited to prevent the network from going toward undesirable predictions, thus increasing the accuracy of the score predictions at each time step. As we show below, TSG is related to stochastic Langevin dynamics in terms of the first-order approximation, and hence, is expected to improve generation quality.

**Connection to Langevin dynamics**   Let $\tilde{t} = t + \Delta t$, where $\Delta t$ is a small perturbation. Using a Taylor expansion, we get $D_{\boldsymbol{\theta}}(\boldsymbol{z}_t, \tilde{t}) = D_{\boldsymbol{\theta}}(\boldsymbol{z}_t, t) + \frac{\partial D_{\boldsymbol{\theta}}(\boldsymbol{z}_t, t)}{\partial t}\Delta t$. Hence, $\hat{D}_{\boldsymbol{\theta}}(\boldsymbol{z}_t, \tilde{t}) = D_{\boldsymbol{\theta}}(\boldsymbol{z}_t, t) + (1 - w_{\text{TSG}})\frac{\partial D_{\boldsymbol{\theta}}(\boldsymbol{z}_t, t)}{\partial t}\Delta t$. Based on Equation (4), the score function is equal to

$$\nabla_{\boldsymbol{z}_t} \log \hat{p}_t(\boldsymbol{z}_t) = \nabla_{\boldsymbol{z}_t} \log p_t(\boldsymbol{z}_t) + \frac{1 - w_{\text{TSG}}}{\sigma(t)^2}\frac{\partial D_{\boldsymbol{\theta}}(\boldsymbol{z}_t, t)}{\partial t}\Delta t. \tag{9}$$

Now, if we follow the Euler sampling step for solving Equation (1), i.e. we define the update rule as $\boldsymbol{z}_{t-1} = \boldsymbol{z}_t + \eta_t \nabla_{\boldsymbol{z}_t} \log \hat{p}_t(\boldsymbol{z}_t)$, then the modified sampling step after time-step guidance will be equal to

$$\boldsymbol{z}_{t-1} = \boldsymbol{z}_t + \eta_t \nabla_{\boldsymbol{z}_t} \log p_t(\boldsymbol{z}_t) + \eta_t \frac{1 - w_{\text{TSG}}}{\sigma(t)^2}\frac{\partial D_{\boldsymbol{\theta}}(\boldsymbol{z}_t, t)}{\partial t}\Delta t. \tag{10}$$

Assuming that $\Delta t$ is a Gaussian random variable with zero mean, the update rule resembles a Langevin dynamics step, where the noise strength is determined based on the network behavior as represented by $\frac{\partial D_{\boldsymbol{\theta}}(\boldsymbol{z}_t, t)}{\partial t}$. As Langevin dynamics is known to increase the quality of sampling from a given distribution by compensating for the errors happening at each sampling step, we argue that TSG also behaves similarly in terms of first-order approximation.

**Implementation details**   In practice, we implement TSG by perturbing the time-step embedding with zero-mean Gaussian noise according to $\tilde{t}_{\text{emb}} = t_{\text{emb}} + st^{\alpha}\boldsymbol{n}$ where $\boldsymbol{n} \sim \mathcal{N}(\boldsymbol{0}, \boldsymbol{I})$ and $st^{\alpha}$ determines the noise scale at each time step $t$. We choose $s$ and $\alpha$ such that the scale of the noise portion becomes comparable to the scale of the time-step embedding $t_{\text{emb}}$. Empirically, we also find that it is sometimes beneficial to apply the perturbed embeddings only to a portion of layers in the diffusion network, e.g., using $\tilde{t}_{\text{emb}}$ for the first 10 layers and $t_{\text{emb}}$ for the rest of layers. We provide ablations on these hyperparameters in Section 7.

## 6   EXPERIMENTS

In this section, we rigorously evaluate ICG and demonstrate its ability to simulate the behavior of CFG across several conditional models. Additionally, we show that TSG improves the quality of both conditional and unconditional generations compared to the non-guided sampling baseline.

**Setup**   All experiments are conducted via pre-trained checkpoints provided by official implementations. We use the recommended sampler that comes with each model, such as the EDM sampler for EDM networks (Karras et al., 2022), DPM++ (Lu et al., 2022b) for Stable Diffusion (Rombach et al., 2022), and DDPM (Ho et al., 2020) for DiT-XL/2 (Peebles & Xie, 2022).

**Evaluation**   We use Fréchet Inception Distance (FID) (Heusel et al., 2017) as the main metric to measure both quality and diversity due to its alignment with human judgment. As FID is known to be sensitive to small implementation details, we ensure that models under comparison follow the same evaluation setup. For completeness, we also report precision (Kynkäänniemi et al., 2019) as a standalone quality metric and recall (Kynkäänniemi et al., 2019) as a diversity metric whenever possible. $\text{FD}_{\text{DINOv2}}$ (Stein et al., 2024) is also reported for the EDM2 model (Karras et al., 2023).

### 6.1   COMPARISON BETWEEN ICG AND CFG

**Qualitative results**   The qualitative comparisons between ICG and CFG are given in Figure 1 for Stable Diffusion (Rombach et al., 2022) and DiT-XL/2 (Peebles & Xie, 2022) models. Figure 2 also shows a comparison between the EDM2 model (Karras et al., 2023) guided with a separate unconditional module vs ICG. We observe that both ICG and CFG improve image quality, and the outputs of ICG and CFG are almost identical. This empirical evidence agrees with our theoretical justification provided in Section 4.

w/o guidance CFG ICG (Ours)

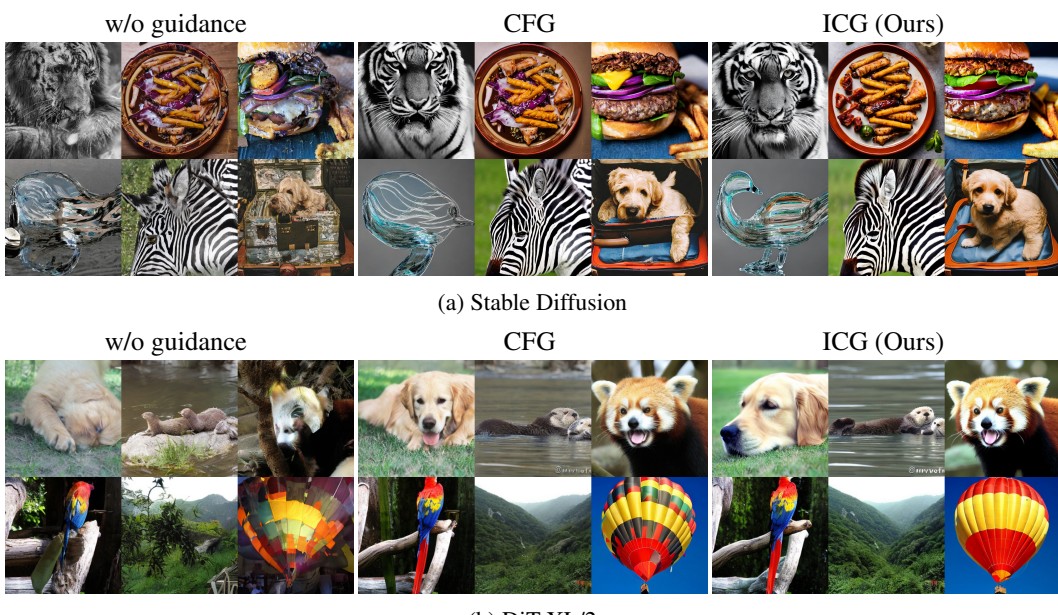

(a) Stable Diffusion

w/o guidance CFG ICG (Ours)

(b) DiT-XL/2

Figure 1: Comparison between CFG and ICG for (a) Stable Diffusion (Rombach et al., 2022) and (b) DiT-XL/2 (Peebles & Xie, 2022). Both CFG and ICG significantly improve the image quality of the baseline. Also note the similarity between the outputs of CFG and ICG, confirming our theoretical analysis in Section 4.

w/o guidance CFG ICG (Ours)

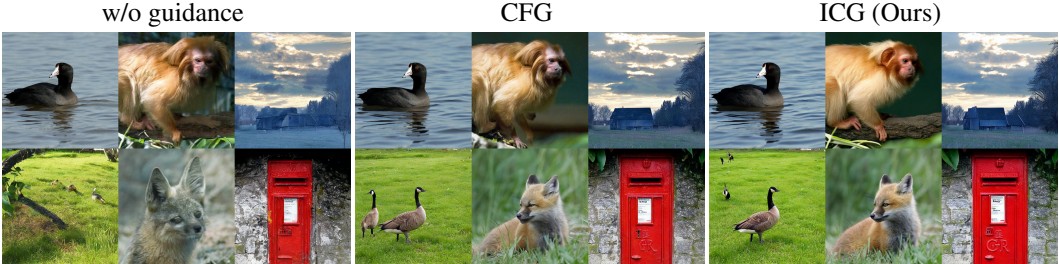

Figure 2: Comparison between the EDM2 model (Karras et al., 2023) guided with another unconditional module and ICG. We observe that using ICG leads to similar generations as CFG, and both methods significantly improve output quality compared to sampling without guidance.

**Quantitative results** We now show that ICG and CFG both result in similar performance metrics across several conditional models. As shown in Table 1, compared to CFG, ICG achieves better or similar performance across all metrics.[2] In Table 2, we also compare the effect of ICG on EDM (Karras et al., 2022) and EDM2 (Karras et al., 2023) models that were not trained with the CFG objective. The table shows that ICG performs similarly to guiding the generations with a separate unconditional module.

**Effect of removing the CFG objective from training** In this experiment, we demonstrate that the training component allocated to the CFG objective (i.e., label dropping) is unnecessary, and better results are obtained by training a purely conditional model and guiding the generations at inference with ICG. Using a DiT model for class-conditional ImageNet generation, Figure 3 shows that the purely conditional model consistently outperforms standard training with label dropping ($p = 0.1$) across all checkpoints. Consequently, training resources can be reallocated to the conditional part, leading to either faster convergence (by approximately 30%) or a better model (with around a 20% reduction in FID) with the same number of training iterations.

---

[2]For MDM (Tevet et al., 2023), recall is not available, and R-precision is reported similar to the paper.

Table 1: Quantitative comparison between CFG and ICG. ICG is able to achieve similar metrics to standard CFG by extracting the unconditional score from the conditional model itself.

| Model | Architecture | Guidance | FID ↓ | Precision ↑ | Recall ↑ |
|---|---|---|---|---|---|
| Stable Diffusion (Rombach et al., 2022) | UNet | CFG | 20.13 | **0.69** | **0.54** |
| | | ICG (Ours) | **20.05** | **0.69** | 0.53 |
| DiT-XL/2 (Peebles & Xie, 2022) | Transformer | CFG | 5.56 | 0.81 | **0.66** |
| | | ICG (Ours) | **5.50** | **0.83** | 0.65 |
| Pose-to-Image (Sadat et al., 2024) | UNet | CFG | 14.61 | 0.93 | 0.02 |
| | | ICG (Ours) | **13.46** | **0.94** | **0.03** |
| MDM (Tevet et al., 2023) | Transformer | CFG | 0.65 | **0.73** | - |
| | | ICG (Ours) | **0.47** | 0.71 | - |

Table 2: Quantitative comparison between CFG and ICG for EDM networks. Although these models are not trained with the CFG objective, guiding their generations using a separate unconditional module results in similar outcomes to using ICG.

| Model | Dataset | Guidance | FID ↓ | FD$_{\text{DINOv2}}$ ↓ |
|---|---|---|---|---|
| EDM2-XS (Karras et al., 2023) | Imagenet | CFG | 3.36 | 79.94 |
| | | ICG (Ours) | **3.35** | **79.54** |
| EDM (Karras et al., 2022) | CIFAR-10 | CFG | **1.87** | - |
| | | ICG (Ours) | **1.87** | - |

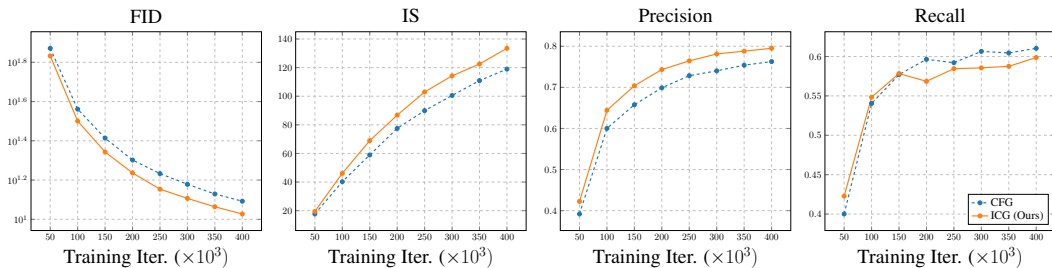

Figure 3: Comparison of CFG and ICG during training of a DiT model on ImageNet. Compared to standard CFG with label dropping, using ICG with a purely conditional model achieves better FID across all checkpoints. This indicates that the iterations spent on the CFG objective could be better allocated to training the conditional score, ultimately leading to a better model.

**Varying the Guidance Scale** Next, we demonstrate that by varying the guidance scale of ICG, we can increase the quality of outputs in a manner similar to standard CFG. As shown in Figure 4, increasing the guidance scale improves precision but reduces recall. The FID plots also form a U-shaped curve, consistent with what we expect from standard CFG.

**Results for ControlNet** We also show that ICG can be used for improving the quality of image-conditioned models as well. We use ControlNet (Zhang & Agrawala, 2023) as an example in this section since it is not trained with the CFG objective on the image condition input. That is, it only applies CFG to the text component of the condition. Our results are given in Figure 5. We see that without any text prompt, ICG significantly improves the quality of generations over the base sampling.

## 6.2 EFFECTIVENESS OF TIME-STEP GUIDANCE

Lastly, we show the effectiveness of time-step guidance in improving generation quality without relying on any information about the conditioning signal. The qualitative results are given in Figure 6. We can see that TSG increases the image quality of both conditional and unconditional sampling. Table 3 also presents the quantitative evaluation of TSG for both conditional and unconditional generation. Similar to CFG, using TSG significantly improves FID by trading diversity with quality. Finally, Figure 7 shows how TSG behaves as we increase the guidance scale. We observe that similar to CFG, TSG also has a U-shaped plot for the FID as the guidance scale increases.

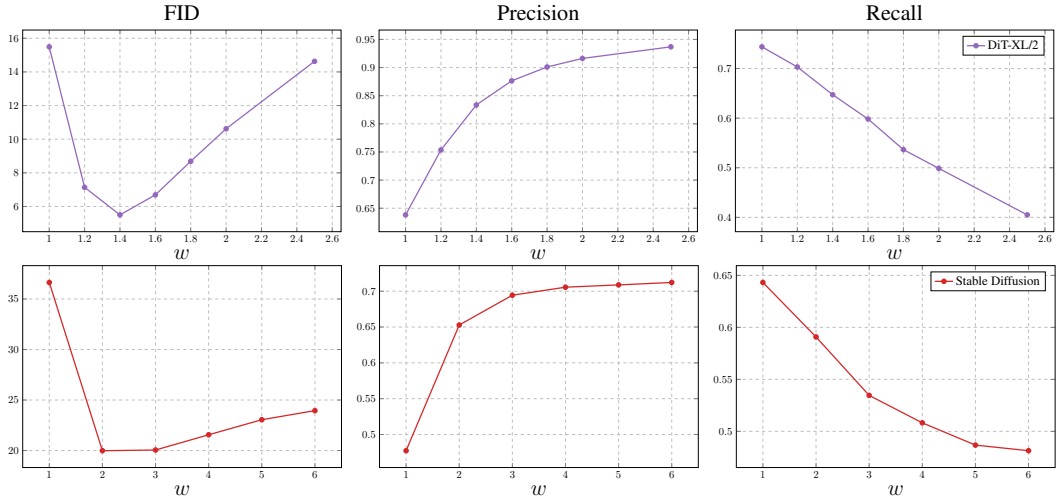

Figure 4: Behavior of ICG as the guidance scale increases. Similar to CFG, ICG trades diversity (lower recall) for quality (higher precision) at higher guidance scales.

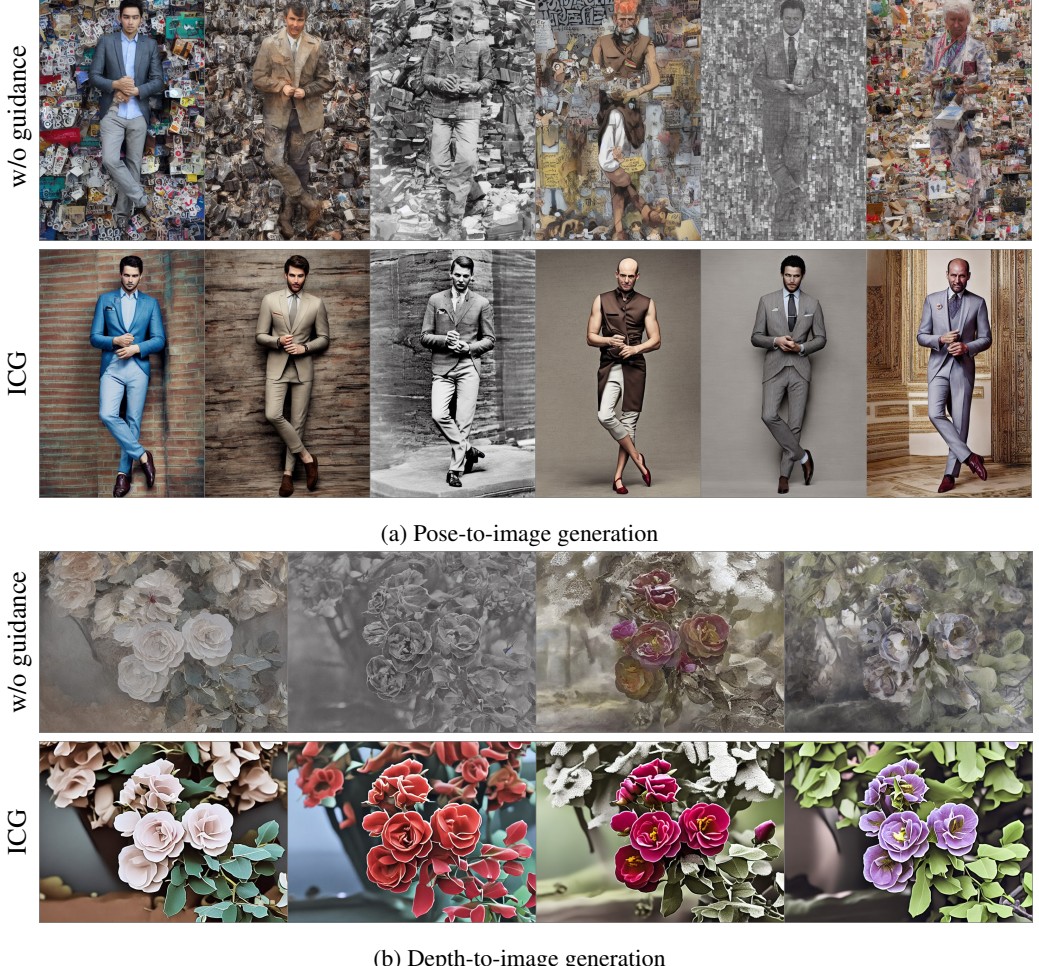

(a) Pose-to-image generation

(b) Depth-to-image generation

Figure 5: Image-conditioned generation with ControlNet (without prompt). ICG significantly increases the quality of generations by applying guidance to the image condition.

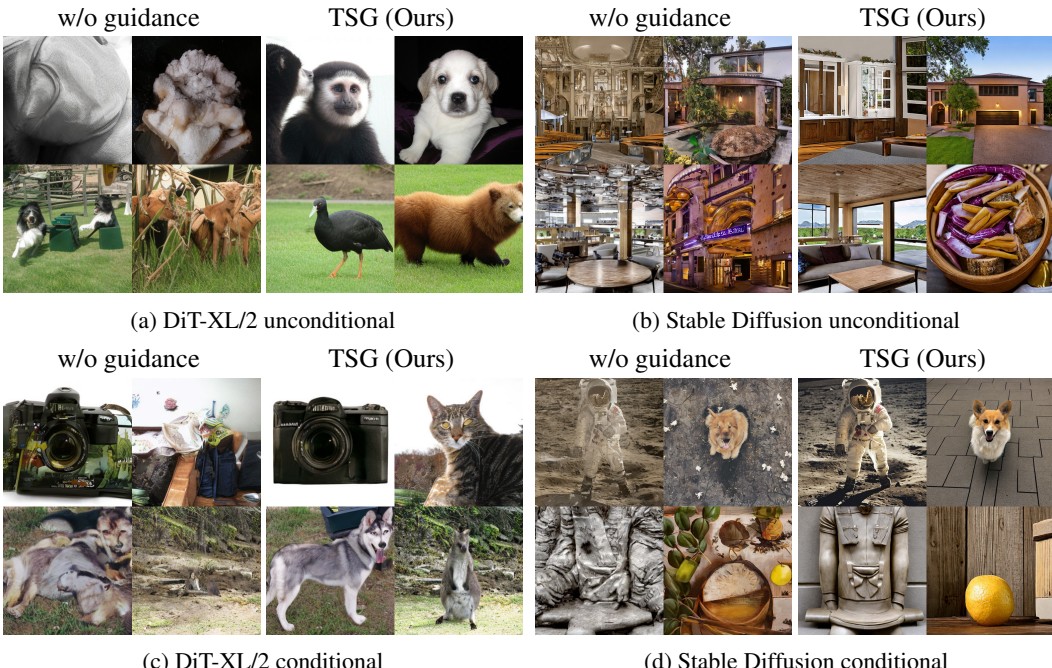

w/o guidance      TSG (Ours)      w/o guidance      TSG (Ours)

(a) DiT-XL/2 unconditional      (b) Stable Diffusion unconditional

w/o guidance      TSG (Ours)      w/o guidance      TSG (Ours)

(c) DiT-XL/2 conditional      (d) Stable Diffusion conditional

Figure 6: Effectiveness of TSG to improve the quality of both unconditional and conditional generation across two different models: DiT-XL/2 (Peebles & Xie, 2022) for class-conditional generation, and Stable Diffusion (Rombach et al., 2022) for text-conditional generation.

Table 3: Quantitative comparison between the baseline sampling of the diffusion models and sampling with TSG. TSG significantly boosts quality (lower FID) across various setups.

| Model | Architecture | Type | Guidance | FID ↓ | Precision ↑ | Recall ↑ |
|---|---|---|---|---|---|---|
| Stable Diffusion (Rombach et al., 2022) | UNet | Unconditional | ✗ | 69.38 | 0.42 | 0.49 |
| | | | TSG (Ours) | **56.65** | **0.54** | **0.54** |
| | | Text-conditional | ✗ | 36.63 | 0.48 | **0.64** |
| | | | TSG (Ours) | **22.17** | **0.62** | 0.59 |
| DiT-XL/2 (Peebles & Xie, 2022) | Transformer | Unconditional | ✗ | 48.67 | 0.48 | **0.59** |
| | | | TSG (Ours) | **29.03** | **0.69** | 0.55 |
| | | Class-conditional | ✗ | 15.49 | 0.64 | **0.74** |
| | | | TSG (Ours) | **6.39** | **0.82** | 0.65 |

Figure 7: Behavior of TSG as the guidance scale increases for DiT-XL/2. Similar to CFG, TSG also significantly improves FID by trading diversity (recall) with quality (precision).

**Combining TSG and ICG** We also demonstrate that ICG and TSG can be complementary to each other when combined at the proper scale. The quantitative results of this experiment are presented in Table 4 with a visual example given in Figure 8. The table indicates that the combination of ICG and TSG outperforms each method in isolation in terms of FID, and all guided sampling algorithms significantly outperform the non-guided baseline.

Table 4: Compatibility of ICG and TSG

| ICG | TSG | FID ↓ | Precision ↑ | Recall ↑ |
|---|---|---|---|---|
| ✗ | ✗ | 15.49 | 0.64 | **0.74** |
| ✓ | ✗ | 6.47 | 0.77 | 0.69 |
| ✗ | ✓ | 9.55 | 0.70 | 0.71 |
| ✓ | ✓ | **5.76** | **0.82** | 0.65 |

| w/o guidance | ICG | ICG + TSG |
|---|---|---|

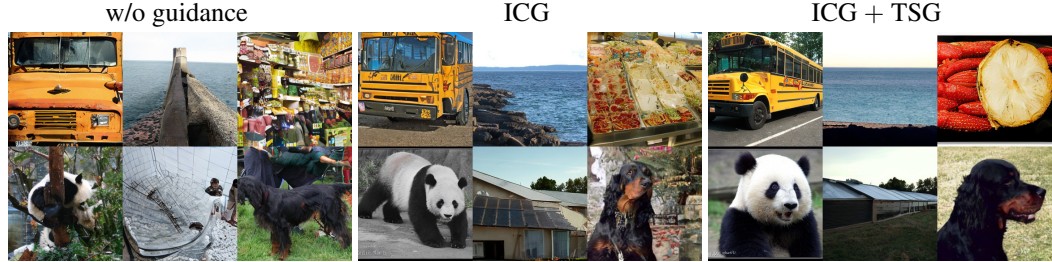

Figure 8: Visual example of combining ICG and TSG. The combination yields better visual generations compared to the baseline and using ICG alone.

Table 6: Ablation study examining various design elements in TSG.

(a) Influence of the noise scale $s$

| $s$ | FID ↓ | Precision ↑ | Recall ↑ |
|---|---|---|---|
| 1.0 | 10.23 | 0.69 | 0.35 |
| 2.0 | **6.85** | **0.80** | **0.69** |
| 2.5 | 7.94 | **0.80** | **0.69** |

(b) Influence of $\alpha$

| $\alpha$ | FID ↓ | Precision ↑ | Recall ↑ |
|---|---|---|---|
| 0.75 | 7.22 | 0.82 | 0.62 |
| 1.0 | **6.39** | **0.84** | 0.65 |
| 1.25 | 6.47 | 0.78 | **0.66** |

(c) Maximum layer index

| Index | FID ↓ | Precision ↑ | Recall ↑ |
|---|---|---|---|
| 5 | 7.84 | 0.75 | **0.69** |
| 10 | **6.85** | 0.79 | 0.65 |
| 15 | 7.65 | **0.82** | 0.65 |

## 7 ABLATION STUDIES

We next present the ablation studies on the effect of random conditioning in ICG and the hyperparameters in TSG. All ablations are conducted using the DiT-XL/2 model (Peebles & Xie, 2022).

**The choice of random condition in ICG** We first show that both Gaussian noise and random conditions can be used for estimating the unconditional part in ICG. The quantitative results are given in Table 5. The table shows that both methods are viable options for simulating classifier-free guidance without training.

Table 5: Ablation on the choice of independent condition in ICG.

| ICG method | FID ↓ | Precision ↑ | Recall ↑ |
|---|---|---|---|
| Gaussian noise | **5.50** | **0.83** | **0.65** |
| Random condition | 5.55 | **0.83** | **0.65** |

**Impact of hyperparameters in time-step guidance** This ablation study explores the effect of hyperparameters in TSG. The results are presented in Table 6. We observe that as we introduce more perturbation into the time-step embedding of the model, in the form of higher noise scale $s$ (Table 6a), lower power $\alpha$ (Table 6b), or higher layer index (Table 6c), precision improves while recall decreases. This suggests that the amount of perturbation should be balanced for a good trade-off between diversity and quality. We also empirically observed that adding too much noise to the time-step embedding hurts image quality.

## 8 DISCUSSION AND CONCLUSION

In this paper, we revisited the core aspects of classifier-free guidance and showed that by replacing the conditional vector in a trained conditional diffusion model with an independent condition, we can efficiently estimate the score of the unconditional distribution. We then introduced independent condition guidance (ICG), a novel method that simulates the same behavior as CFG without the need to learn an unconditional model during training. Inspired by this, we also proposed time-step guidance (TSG) and demonstrated that the time-step information learned by the diffusion model can be leveraged to enhance the quality of generations, even for unconditional models. Our experiments indicate that ICG performs similarly to standard CFG and alleviates the need to consider the CFG objective during training. Thus, ICG streamlines the training of conditional models and improves training efficiency. Additionally, we verified that TSG also improves generation quality in a manner similar to CFG, without relying on any conditional information. As with CFG, challenges remain in accelerating the proposed methods to narrow the gap between the cost of guided and unguided sampling (i.e., eliminating the need to query the diffusion model twice per sampling step); we view this topic as a promising avenue for further research.

ETHICS STATEMENT

As generative modeling advances, the creation and spread of fabricated or inaccurate data become easier. Thus, while improvements in AI-generated content can boost productivity and creativity, it is crucial to consider the associated risks and ethical implications. For a more detailed discussion on the ethics and creativity in computer vision, we refer readers to Rostamzadeh et al. (2021).

REPRODUCIBILITY STATEMENT

This work is based on the official implementations of the pretrained models referenced in the main text. The exact algorithms for ICG and TSG are provided in Algorithms 1 and 2, with corresponding pseudocode shown in Figures 11 and 12. Additional implementation details, including the specific hyperparameters used to generate the results in this paper, are discussed in Appendix G.

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

## A  ANALYSIS OF THE ERROR IN ESTIMATING THE UNCONDITIONAL SCORE

In this section, we provide another perspective on why training the conditional score is sufficient for computing the unconditional score. For ease of exposition, assume that the model's objective is to directly learn the conditional score, $s_\theta(z, y) \approx \nabla_z \log p(z|y)$, from paired data jointly drawn from $p(z, y)$. This can be achieved by directly using denoising score matching (Song et al., 2021b) or by training a denoiser $D_\theta(z, y)$ and converting its outputs to the corresponding score function via Equation (4). The question then boils down to whether the unconditional score, $\nabla_z \log p(z)$, can be recovered from the conditional score.

**Lemma A.1.** *The unconditional score, $\nabla_z \log p(z)$, is equal to the conditional expectation of the conditional score, $\nabla_z \log p(z|y)$.*

*Proof.* By direct calculation, we have that

$$\nabla_z \log p(z) = \frac{\nabla_z p(z)}{p(z)} \tag{11}$$

$$= \frac{\nabla_z \int p(y) p(z|y) \mathrm{d}y}{p(z)} \tag{12}$$

$$= \int \frac{p(y)}{p(z)} \nabla_z p(z|y) \mathrm{d}y \tag{13}$$

$$= \int \frac{p(y) p(z|y)}{p(z) p(z|y)} \nabla_z p(z|y) \mathrm{d}y \tag{14}$$

$$= \int p(y|z) \nabla_z \log p(z|y) \mathrm{d}y. \tag{15}$$

The last term is the conditional expectation of the conditional score, proving the result.  □

This shows that under sufficient data and optimization, the unconditional score at each time step is theoretically available from the conditional score through (conditional) marginalization. It is also therefore the case that a condition drawn from $p(y|z)$ will produce an unbiased estimate of the unconditional score.

We propose drawing conditions randomly from a distribution $q(y)$, which is not necessarily equal to $p(y|z)$. Below we characterize the approximation error incurred by this choice.

**Theorem A.2.** *When the expectation is taken over the distribution $q(y)$, we have*
$$\mathbb{E}_{q(y)}[\nabla_z \log p(z|y)] = \nabla_z \log p(z) - \nabla_z \mathcal{D}_{KL}(q(y) \| p(y|z)).$$
*That is, the approximation error is bounded by the gradient of the KL divergence between $q(y)$ and $p(y|z)$.*

*Proof.* We can write
$$\nabla_z \log p(z|y) = \nabla_z \log p(z, y) - \nabla_z \log p(y) = \nabla_z \log p(z, y),$$
which in turn can be written as
$$\nabla_z \log p(z, y) = \nabla_z \log p(z) + \nabla_z \log p(y|z).$$
Taking the expectation over $q(y)$, we have
$$\mathbb{E}_{q(y)}[\nabla_z \log p(z, y)] = \nabla_z \log p(z) + \mathbb{E}_{q(y)}[\nabla_z \log p(y|z)]$$
$$= \nabla_z \log p(z) + \text{Error}.$$

Now consider the KL divergence between $q(y)$ and $p(y|z)$:
$$\mathcal{D}_{KL}(q(y) \| p(y|z)) = \int q(y)[\log q(y) - \log p(y|z)].$$
Taking the gradient with respect to $z$, we have
$$\nabla_z \mathcal{D}_{KL}(q(y) \| p(y|z)) = - \int q(y) \nabla_z \log p(y|z) \mathrm{d}y = -\mathbb{E}_{q(y)}[\nabla_z \log p(y|z)],$$
which is the negative of the error term above, proving the result.  □

## B   ANALYSIS OF THE ICG ESTIMATOR FOR MIXTURES OF GAUSSIANS

We now analyze the ICG estimator using a mixture of Gaussian distributions so that the ground truth scores are analytically available. Assume we have the data distribution given by $p_{\text{data}}(\boldsymbol{x}) = 0.5\mathcal{N}(\boldsymbol{x}; \boldsymbol{\mu}_0, \boldsymbol{I}) + 0.5\mathcal{N}(\boldsymbol{x}; \boldsymbol{\mu}_1, \boldsymbol{I})$. This gives us $p_{\text{data}}(\boldsymbol{x}|y=0) := p_0(\boldsymbol{x}) = \mathcal{N}(\boldsymbol{x}; \boldsymbol{\mu}_0, \boldsymbol{I})$ and $p_{\text{data}}(\boldsymbol{x}|y=1) := p_1(\boldsymbol{x}) = \mathcal{N}(\boldsymbol{x}; \boldsymbol{\mu}_1, \boldsymbol{I})$. Given the forward process $\boldsymbol{z}_t = \boldsymbol{x} + \sigma_t\boldsymbol{\epsilon}$, we have $p_0(\boldsymbol{z}_t) = \mathcal{N}(\boldsymbol{z}_t; \boldsymbol{\mu}_0, (1+\sigma_t^2)\boldsymbol{I})$ and $p_1(\boldsymbol{z}_t) = \mathcal{N}(\boldsymbol{z}_t; \boldsymbol{\mu}_1, (1+\sigma_t^2)\boldsymbol{I})$. Let $s(\boldsymbol{z}_t, y)$ be the conditional score function $\nabla_{\boldsymbol{z}_t} \log p_t(\boldsymbol{z}_t|y)$ In this case, the conditional score functions are given by

$$s(\boldsymbol{z}_t, y=0) = \frac{\boldsymbol{\mu}_0 - \boldsymbol{x}}{1 + \sigma_t^2}, \quad \text{and} \quad s(\boldsymbol{z}_t, y=1) = \frac{\boldsymbol{\mu}_1 - \boldsymbol{x}}{1 + \sigma_t^2}. \tag{16}$$

The unconditional score function is equal to

$$s(\boldsymbol{z}_t) = \frac{0.5\,\nabla_{\boldsymbol{z}_t}\,p_0(\boldsymbol{z}_t) + 0.5\,\nabla_{\boldsymbol{z}_t}\,p_1(\boldsymbol{z}_t)}{0.5 p_0(\boldsymbol{z}_t) + 0.5 p_1(\boldsymbol{z}_t)} \tag{17}$$

$$= \frac{\nabla_{\boldsymbol{z}_t}\,\mathcal{N}(\boldsymbol{x}; \boldsymbol{\mu}_0, (1+\sigma_t^2)\boldsymbol{I}) + \nabla_{\boldsymbol{z}_t}\,\mathcal{N}(\boldsymbol{x}; \boldsymbol{\mu}_1, (1+\sigma_t^2)\boldsymbol{I})}{p_0(\boldsymbol{z}_t) + p_1(\boldsymbol{z}_t)} \tag{18}$$

$$= \frac{p_0(\boldsymbol{z}_t)}{p_0(\boldsymbol{z}_t) + p_1(\boldsymbol{z}_t)}\frac{\boldsymbol{\mu}_0 - \boldsymbol{x}}{1 + \sigma_t^2} + \frac{p_1(\boldsymbol{z}_t)}{p_0(\boldsymbol{z}_t) + p_1(\boldsymbol{z}_t)}\frac{\boldsymbol{\mu}_1 - \boldsymbol{x}}{1 + \sigma_t^2} \tag{19}$$

The CFG update direction is therefore given by

$$s(\boldsymbol{z}_t, y=0) - s(\boldsymbol{z}_t) = \frac{\boldsymbol{\mu}_0 - \boldsymbol{x}}{1 + \sigma_t^2} - \frac{p_0(\boldsymbol{z}_t)}{p_0(\boldsymbol{z}_t) + p_1(\boldsymbol{z}_t)}\frac{\boldsymbol{\mu}_0 - \boldsymbol{x}}{1 + \sigma_t^2} - \frac{p_1(\boldsymbol{z}_t)}{p_0(\boldsymbol{z}_t) + p_1(\boldsymbol{z}_t)}\frac{\boldsymbol{\mu}_1 - \boldsymbol{x}}{1 + \sigma_t^2} \tag{20}$$

$$= \frac{p_1(\boldsymbol{z}_t)}{p_0(\boldsymbol{z}_t) + p_1(\boldsymbol{z}_t)}\frac{\boldsymbol{\mu}_0 - \boldsymbol{x}}{1 + \sigma_t^2} - \frac{p_1(\boldsymbol{z}_t)}{p_0(\boldsymbol{z}_t) + p_1(\boldsymbol{z}_t)}\frac{\boldsymbol{\mu}_1 - \boldsymbol{x}}{1 + \sigma_t^2} \tag{21}$$

$$= \frac{p_1(\boldsymbol{z}_t)}{p_0(\boldsymbol{z}_t) + p_1(\boldsymbol{z}_t)}\frac{\boldsymbol{\mu}_0 - \boldsymbol{\mu}_1}{1 + \sigma_t^2} \tag{22}$$

For ICG, assume that the random condition $\hat{y}$ is sampled from $q(\hat{y})$. Therefore, the update rule given by ICG is equal to

$$s(\boldsymbol{z}_t, y=0) - s(\boldsymbol{z}_t, \hat{y}) = \begin{cases} 0 & \hat{y} = 0, \\ \frac{\boldsymbol{\mu}_0 - \boldsymbol{\mu}_1}{1 + \sigma_t^2} & \hat{y} = 1. \end{cases} \tag{23}$$

This means that on expectation, the update direction of ICG is equal to

$$\mathbb{E}_{q(\hat{y})}[s(\boldsymbol{z}_t, y=0) - s(\boldsymbol{z}_t, \hat{y})] = q(\hat{y}=1)\frac{\boldsymbol{\mu}_0 - \boldsymbol{\mu}_1}{1 + \sigma_t^2}. \tag{24}$$

This example shows that in theory, we can set $q(\hat{y} = 1) = \frac{p_1(\boldsymbol{z}_t)}{p_0(\boldsymbol{z}_t) + p_1(\boldsymbol{z}_t)}$, and the ICG estimator becomes an unbiased estimate of the CFG update direction. In practice and for real-world models, we noted that it is sufficient to set $q(\hat{y})$ to either a uniform distribution over the space of the conditions, or equal to a Gaussian distribution with suitable standard deviation.

## C   COMPATIBILITY OF ICG WITH CADS

We show that ICG is compatible with CADS (Sadat et al., 2024), and CADS can be used on top of ICG to increase the diversity of generations. An example of applying CADS to ICG is shown in Figure 9, and the quantitative results are given in Table 7 for the DiT-XL/2 model. As ICG behaves similarly to the standard CFG, applying CADS increases diversity with minimal drop in quality.

Table 7: Effectiveness of CADS on ICG.

| Guidance | FID ↓ | Precision ↑ | Recall ↑ |
|---|---|---|---|
| ICG | 20.56 | **0.89** | 0.32 |
| +CADS | **8.83** | 0.78 | **0.61** |

## D   INTUITION BEHIND TSG

This section provides more intuition on time-step guidance. We demonstrate that if we perturb the time step with a positive or negative constant (using $t + \delta$ or $t - \delta$) to guide the sampling, it results

|  ICG  |  +CADS  |  ICG  |  +CADS  |

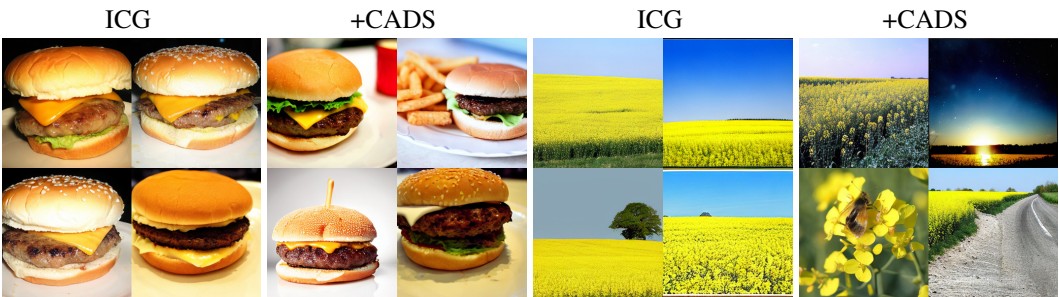

Figure 9: Similar to CFG, ICG is compatible with CADS, and CADS can be used to increase the diversity of ICG at higher guidance scales. Samples are generated from the DiT-XL/2 model.

|  $t - \delta$  |  $t + \delta$  |  TSG  |

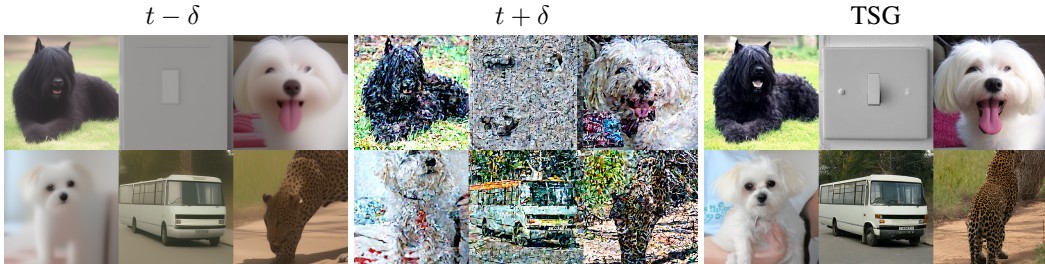

Figure 10: Intuition behind TSG: Using lower time steps for guidance causes excessive noise removal (soft outputs), while higher time steps cause insufficient noise removal (noisy images). TSG employs both directions to improve output quality.

Table 8: Comparison between TSG and other guidance methods. TSG achieves better quality compared to SAG and PAG while requiring no specific assumption about the underlying architecture of the diffusion model.

| Guidance | FID | Precision | Recall |
|---|---|---|---|
| Unguided | 69.38 | 0.42 | 0.49 |
| SAG (Hong et al., 2022) | 59.34 | 0.48 | 0.53 |
| PAG (Ahn et al., 2024) | 62.85 | 0.42 | 0.51 |
| TSG (ours) | **56.65** | **0.54** | **0.54** |

in insufficient or excessive noise removal in final generations. As shown in Figure 10, using lower time steps causes the model to perform excessive noise removal (soft outputs), while using higher time steps forces the model to perform insufficient noise removal (noisy images). TSG uses both directions to prevent the outputs from moving toward these undesirable paths, thereby increasing the quality of the generations.

# E   COMPARISON BETWEEN TSG AND OTHER METHODS

In this section, we compare the performance of TSG with other guidance mechanisms that improve the generation quality of diffusion models without relying on any condition. We use self-attention guidance (SAG) (Hong et al., 2022), and perturbed-attention guidance (PAG) (Ahn et al., 2024) as two representative methods in this category. The results are given in Table 8. We use Stable Diffusion 2.1 for this parameters and the default hyperparameters for all methods. We note that TSG is more effective in improving the quality of generations, providing better FID compared to SAG and PAG.

Table 9: Comparison between unguided sampling and TSG using different number of sampling steps. TSG is able to achieve significantly better FID compared to both unguided baselines.

| Guidance | # Steps | FID | Precision | Recall |
|---|---|---|---|---|
| Unguided | 100 | 15.49 | 0.6382 | **0.7437** |
| Unguided | 200 | 12.94 | 0.6683 | 0.7387 |
| TSG | 100 | **6.39** | **0.8198** | 0.6489 |

## F    INCREASING THE NUMBER OF SAMPLING STEPS

Since TSG increases the number of sampling steps (similar to CFG), a natural question is whether the same behavior can be achieved by simply increasing the number of sampling steps in the unguided sampling baseline. Our results in Table 9 indicate that this approach performs significantly worse than TSG, suggesting that, like CFG, TSG alters the sampling trajectories toward higher-quality generations. Note that TSG achieves a significantly better FID compared to both unguided sampling baselines. This aligns with findings from the CFG literature, where guided sampling outperforms unguided baselines even with many sampling steps (e.g., 1000 steps) (Dhariwal & Nichol, 2021; Ho & Salimans, 2022).

## G    IMPLEMENTATION DETAILS

The sampling details for ICG and TSG are provided in Algorithms 1 and 2. Both algorithms are straightforward to implement and require minimal code changes compared to the base sampling. The pseudocode for implementing ICG and TSG is also included in Figures 11 and 12. Additionally, the hyperparameters used in our experiments are listed in Tables 10 and 11. The CADS experiment was conducted with a linear schedule using $\tau_1 = 0.5$, $\tau_2 = 0.9$, and $s_{\mathrm{CADS}} = 0.15$. Lastly, please note that we did not perform an exhaustive grid search on the parameters of TSG, and better configurations are likely to exist for each model.

For the TSG noise schedule, we experimented with a constant and a power schedule, as shown in Figure 12, and found that both work similarly. We recommend using the power schedule as it offers more flexibility over the scale of the noise at each $t$. The constant schedule is technically a special case of the power schedule, where the exponent is zero. We also found it useful to apply TSG only at intervals during the sampling, i.e., for $t \in [T_{\min}, T_{\max}]$, where $T_{\min}$ and $T_{\max}$ are hyperparameters. Also, when limiting TSG to only a portion of layers in the diffusion model, we used the first $N$ layers of transformer-based architectures and the first $N$ layers of the encoder and decoder in UNet-based architectures. We chose to scale the amount of noise $s$ based on the standard deviation of the time-step embedding (see Figures 11 and 12) for more fine-grained control over the scale.

We primarily use the ADM evaluation script (Dhariwal & Nichol, 2021) for computing FID, precision, and recall to ensure a fair comparison across experiments. For class-conditional models, the FID is computed between 10,000 (for DiT-XL/2) or 50,000 (For EDM and EDM2) generated images and the full training dataset. For text-to-image models, we use the evaluation subset of MS COCO 2017 (Lin et al., 2014) as the ground truth for captions and images.

## H    MORE VISUAL RESULTS

This section presents additional visual results for our guidance methods. More results on ICG are provided in Figure 13, while additional results for TSG are shown in Figures 14 and 15. Consistent with the main results of the paper, ICG produces similar outcomes to CFG, and TSG consistently enhances the quality compared to the baseline. Figure 16 provides examples of the effectiveness of TSG based on Stable Diffusion XL (SDXL) (Podell et al., 2023). Finally, Figure 17 shows a qualitative comparison between unguided sampling and sampling with TSG for several latent diffusion models from Rombach et al. (2022).

---

**Algorithm 1** Sampling with ICG

---

**Require:** $w_{\text{ICG}}$: ICG strength
**Require:** $\boldsymbol{y}$: Input condition
1: Initial value: $\boldsymbol{z}_1 \sim \mathcal{N}(\boldsymbol{0}, \boldsymbol{I})$
2: **for** $t = T, \dots, 1$ **do**

3:   ∘ Pick a random $\hat{\boldsymbol{y}}$ independent of $\boldsymbol{z}_t$ (Gaussian noise or from the conditioning space).

4:   ∘ Compute the ICG guided output at $t$:
      $\hat{D}_{\text{ICG}}(\boldsymbol{z}_t, t, \boldsymbol{y}) = D(\boldsymbol{z}_t, t, \hat{\boldsymbol{y}}) + w_{\text{ICG}}(D(\boldsymbol{z}_t, t, \boldsymbol{y}) - D(\boldsymbol{z}_t, t, \hat{\boldsymbol{y}}))$.

5:   ∘ Perform one sampling step (e.g. one step of DDPM):
      $\boldsymbol{z}_{t-1} = \texttt{diffusion\_reverse}(\hat{D}_{\text{ICG}}, \boldsymbol{z}_t, t)$.

6: **end for**
7: **return** $\boldsymbol{z_0}$

---

**Algorithm 2** Sampling with TSG

---

**Require:** $w_{\text{TSG}}$: TSG strength
**Require:** $(s, \alpha)$: TSG hyperparameters
**Require:** $\boldsymbol{y}$: Input condition (optional)
1: Initial value: $\boldsymbol{z}_1 \sim \mathcal{N}(\boldsymbol{0}, \boldsymbol{I})$
2: **for** $t = T, \dots, 1$ **do**

3:   ∘ Perturb the time-step embedding $t_{\text{emb}}$ to get $\hat{t}_{\text{emb}}$:
      $\hat{t}_{\text{emb}} = t_{\text{emb}} + st^{\alpha}\boldsymbol{n}$, where $\boldsymbol{n} \sim \mathcal{N}(\boldsymbol{0}, \boldsymbol{I})$.

4:   ∘ Compute the TSG guided output at $t$:
      $\hat{D}_{\text{TSG}}(\boldsymbol{z}_t, t, \boldsymbol{y}) = D(\boldsymbol{z}_t, \hat{t}_{\text{emb}}, \boldsymbol{y}) + w_{\text{TSG}}(D(\boldsymbol{z}_t, t_{\text{emb}}, \boldsymbol{y}) - D(\boldsymbol{z}_t, \hat{t}_{\text{emb}}, \boldsymbol{y}))$.

5:   ∘ Perform one sampling step (e.g. one step of DDPM):
      $\boldsymbol{z}_{t-1} = \texttt{diffusion\_reverse}(\hat{D}_{\text{TSG}}, \boldsymbol{z}_t, t)$.

6: **end for**
7: **return** $\boldsymbol{z_0}$

---

Table 10: Hyperparameters used for the ICG experiments.

| Model | ICG mode | ICG scale | CFG scale |
|---|---|---|---|
| DiT-XL/2 | Random class | 1.4 | 1.5 |
| Stable Diffusion | Random text | 3.0 | 4.0 |
| Pose-to-Image | Gaussian noise | 3.0 | 4.0 |
| MDM | Gaussian noise | 2.5 | 2.5 |
| EDM | Random class | 1.05 | 1.1 |
| EDM2 | Random class | 1.25 | 1.25 |

Table 11: Hyperparameters used for the TSG experiments.

| Model | Mode | TSG function | TSG scale | TSG parameters |
|---|---|---|---|---|
| DiT-XL/2 | Unconditional | `constant_schedule` | 5.0 | $\texttt{T\_MIN} = 200, \texttt{T\_MAX} = 800, s = 1.0$ |
| DiT-XL/2 | Conditional | `power_schedule` | 2.5 | $\texttt{T\_MIN} = 0, \texttt{T\_MAX} = 1000, \alpha = 1, s = 2$ |
| Stable Diffusion | Unconditional | `constant_schedule` | 3.0 | $\texttt{T\_MIN} = 100, \texttt{T\_MAX} = 900, s = 1.25$ |
| Stable Diffusion | Conditional | `power_schedule` | 4.0 | $\texttt{T\_MIN} = 400, \texttt{T\_MAX} = 1000, s = 3.0, \alpha = 0.25$ |

```python
1   def get_random_class():
2       """Random class labels."""
3       y_random = torch.randint(0, NUM_CLASSES, (BATCH_SIZE, ))
4       return y_random
5
6   def get_random_text():
7       """Random text tokens."""
8       random_idx = torch.randint(0, NUM_TOKENS, (BATCH_SIZE, MAX_LENGTH))
9       random_tokens = text_encoder(random_idx, attention_mask=None)[0]
10      return random_tokens
11
12  def get_gaussian_noise_embedding(embeddings):
13      """Random embedding based on Gaussian noise."""
14      noise_embedding = torch.randn_like(embeddings) * embeddings.std()
15      return noise_embedding
16
17  def get_gaussian_noise_image(image_cond):
18      """Random condition for image-conditional models."""
19      noise_embedding = torch.randn_like(image_cond) * SCALE
20      return noise_embedding
21
```

Figure 11: Implementation details for ICG. The figure presents pseudocode for implementing the random class, random text, and Gaussian noise embedding for the unconditional component in ICG.

```python
1   def get_constant_schedule(t_emb, t, std_scaling=True):
2       """Applies TSG for a portion of sampling (t in [T_MIN, T_MAX])."""
3       if t < T_MIN or t > T_MAX:
4           return t_emb
5
6       noise_scale = S
7       if std_scaling:
8           noise_scale = S * t_emb.std()
9       that_emb = t_emb + torch.randn_like(t_emb) * noise_scale
10      return that_emb
11
12
13  def get_power_schedule(t_emb, t, std_scaling=True):
14      """Applies TSG according to the power schedule."""
15      if t < T_MIN or t > T_MAX:
16          return t_emb
17      noise_scale = S * t ** (ALPHA)
18      if std_scaling:
19          noise_scale = noise_scale * t_emb.std()
20      that_emb = t_emb + torch.randn_like(t_emb) * noise_scale
21      return that_emb
22
```

Figure 12: Implementation details for TSG. We provide two scheduling techniques for perturbing the time-step embedding. We empirically found that both methods perform similarly.

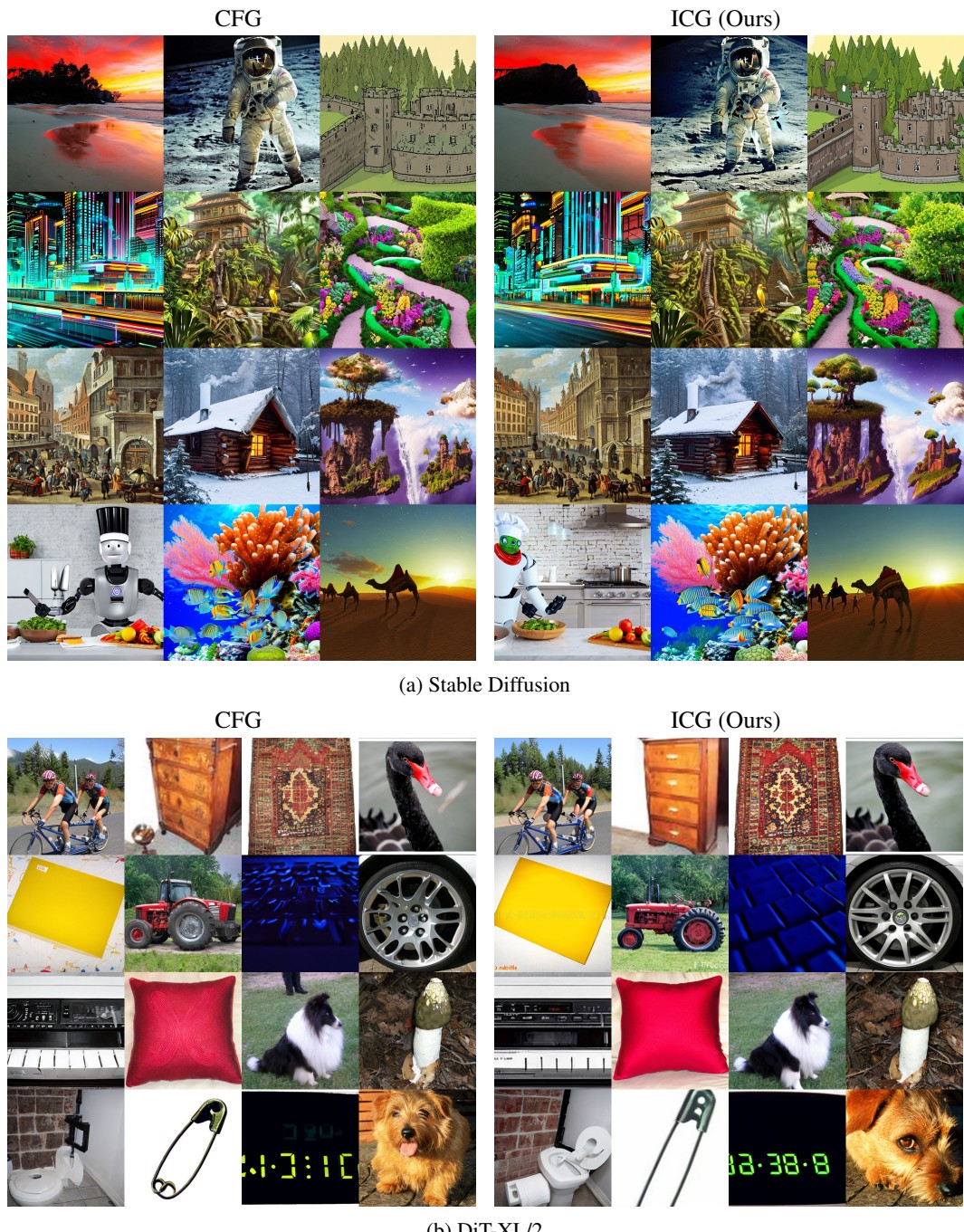

Figure 13: More comparisons between ICG and CFG for (a) text-to-image generation with Stable Diffusion (Rombach et al., 2022) and (b) class-conditional generation with DiT-XL/2 (Peebles & Xie, 2022).

w/o guidance                                    TSG (Ours)

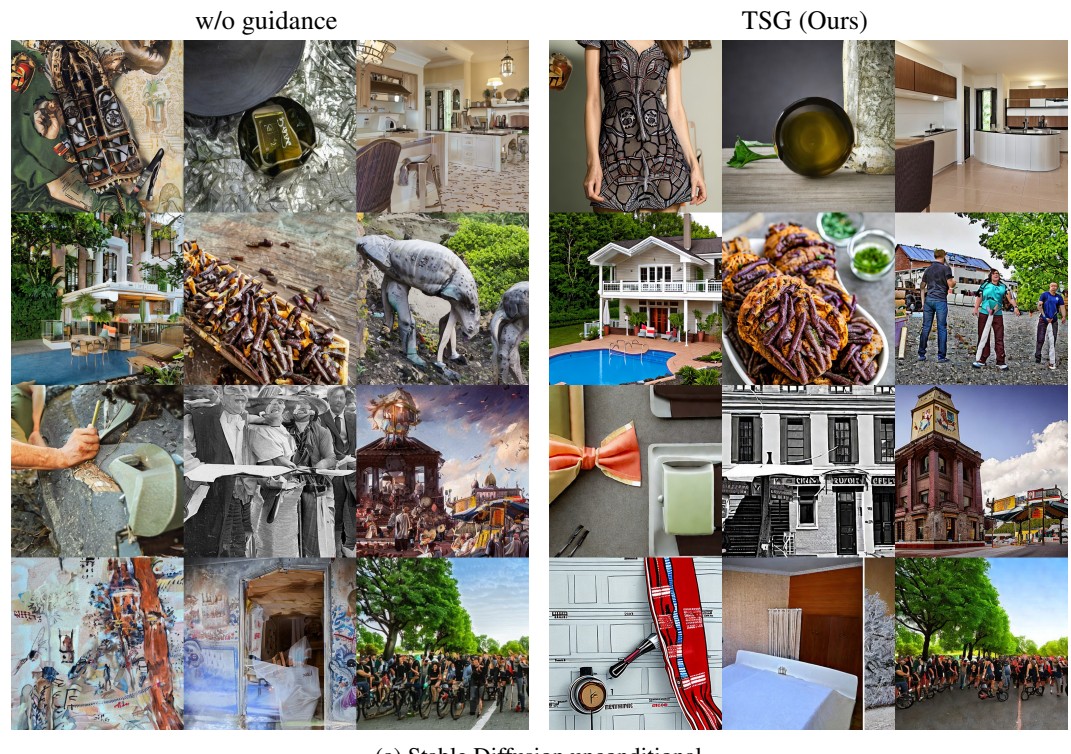

(a) Stable Diffusion unconditional

w/o guidance                                    TSG (Ours)

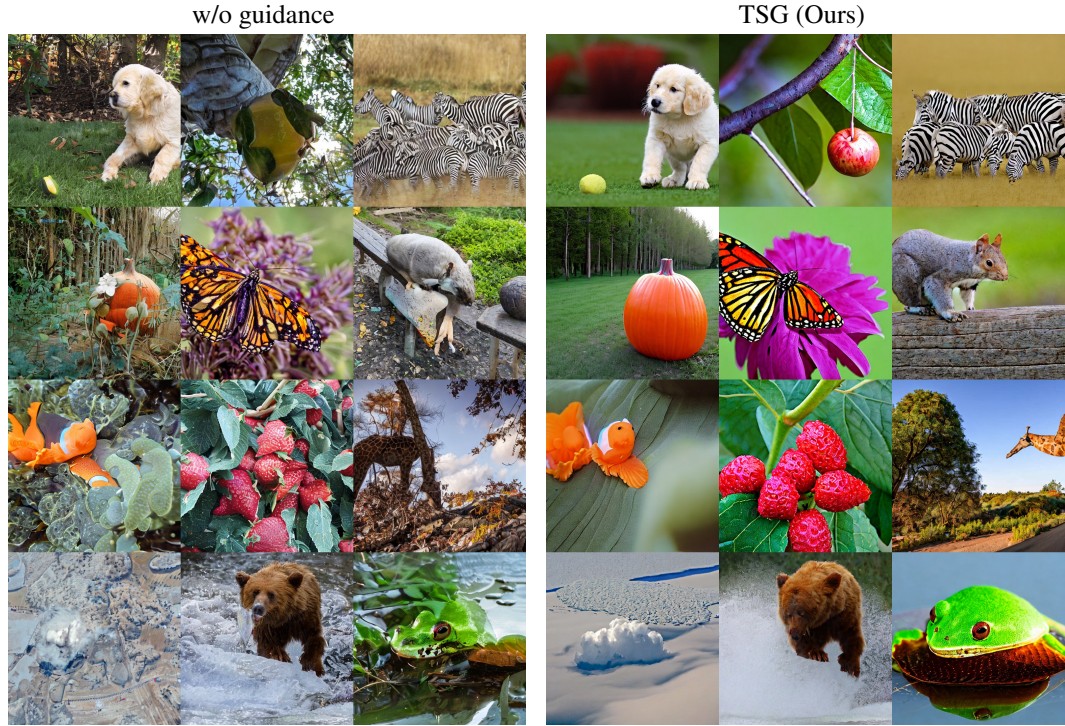

(b) Stable Diffusion conditional

Figure 14: More comparisons on the effectiveness of TSG for improving the quality of both unconditional and conditional generation for Stable Diffusion (Rombach et al., 2022).

w/o guidance                    TSG (Ours)

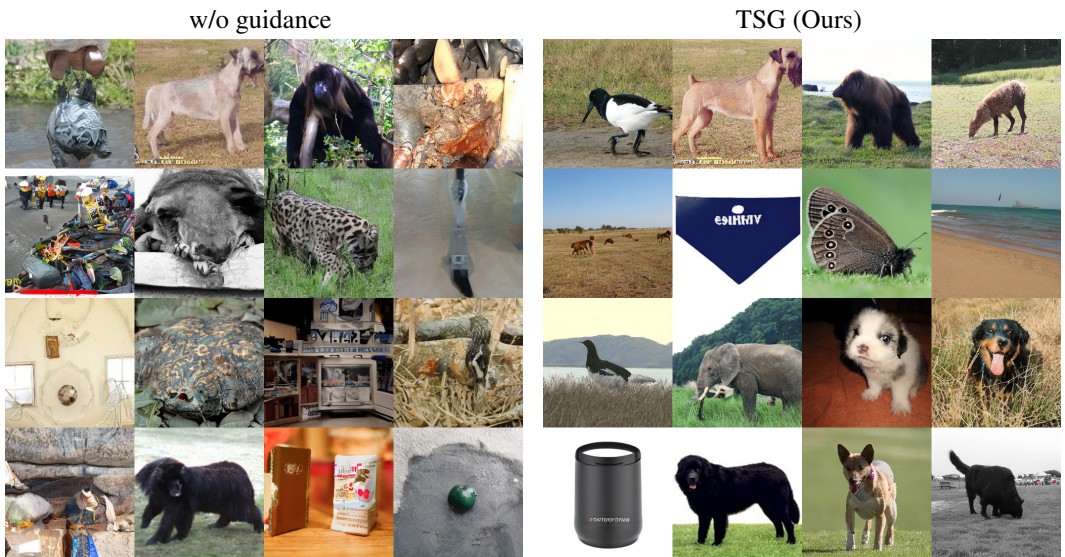

(a) DiT-XL/2 unconditional

w/o guidance                    TSG (Ours)

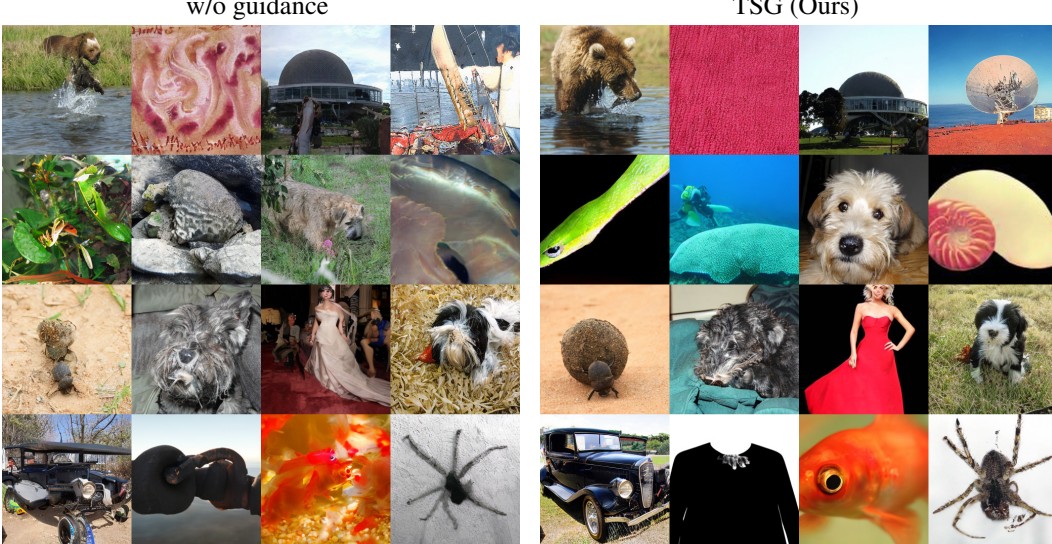

(b) DiT-XL/2 conditional

Figure 15: More comparisons on the effectiveness of TSG for improving the quality of both unconditional and conditional generation for DiT-XL/2 (Peebles & Xie, 2022).

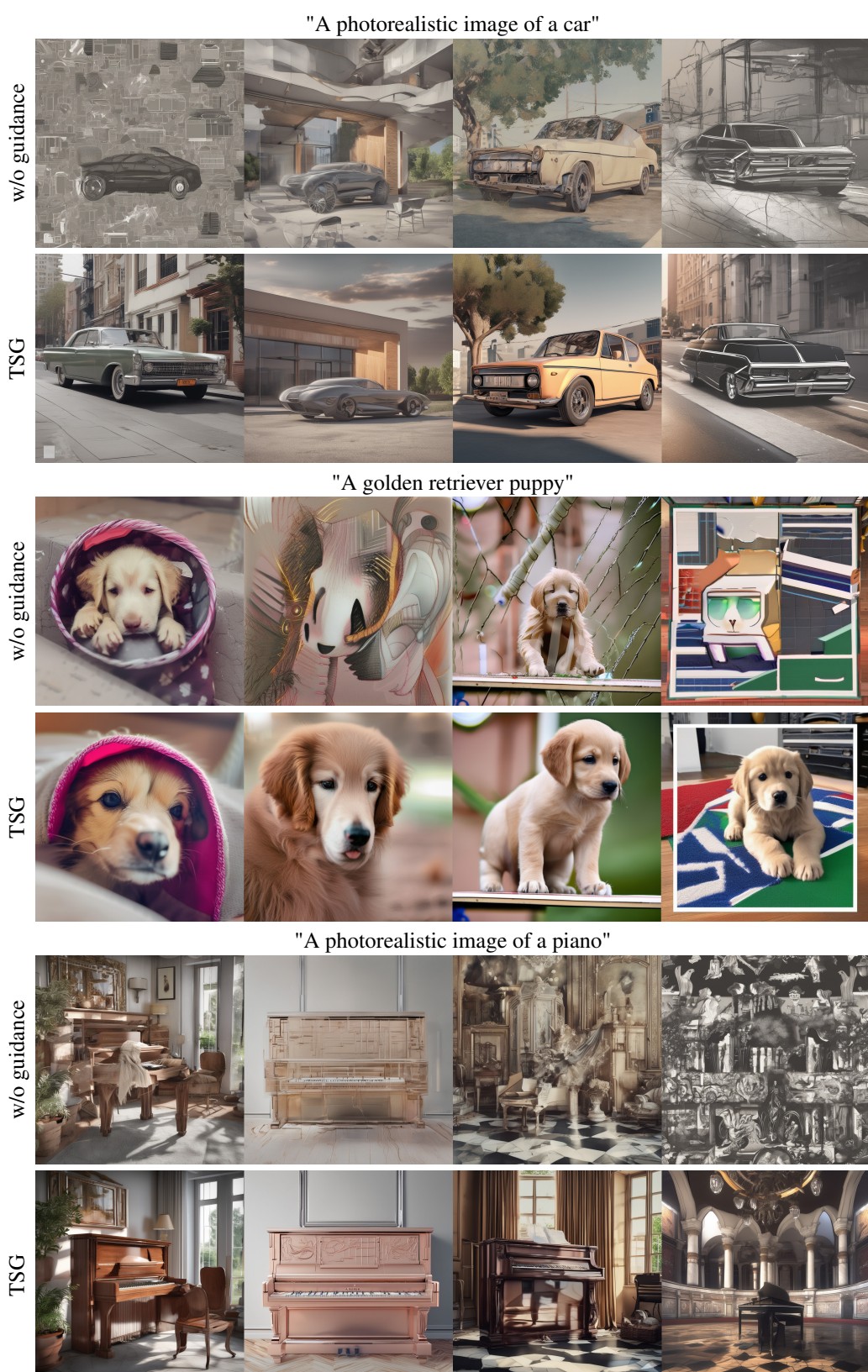

Figure 16: Showcasing the effectiveness of TSG in improving the quality of generations compared to sampling without guidance based on SDXL (Podell et al., 2023).

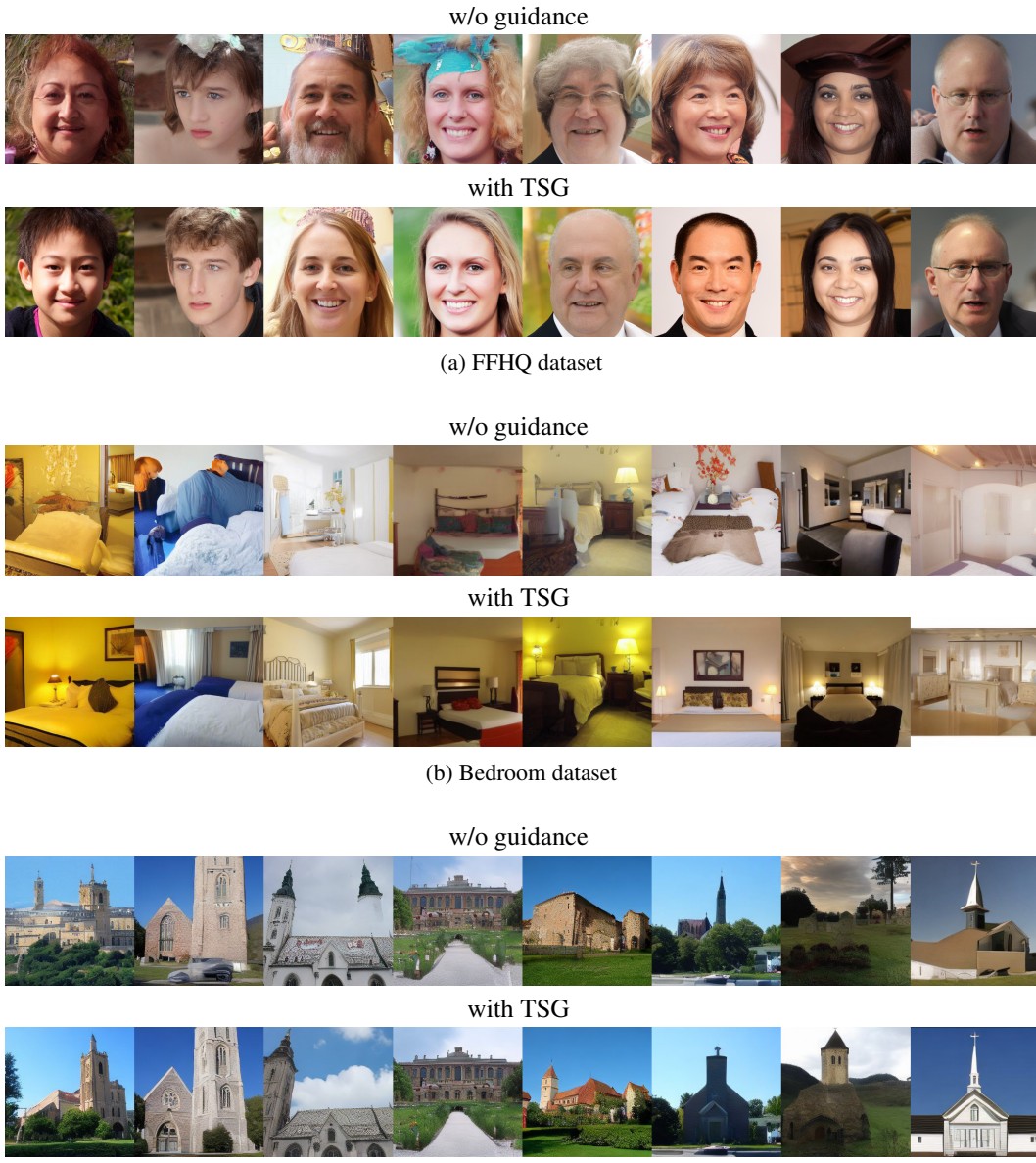

Figure 17: Visual comparison between unguided sampling and sampling with TSG for several unconditional latent diffusion models from Rombach et al. (2022). We observe that TSG consistently improves the quality of all models compared to the baseline sampling.

