# OpenReview forum: "No Training, No Problem: Rethinking Classifier-Free Guidance for Diffusion Models"
_ICLR.cc/2025/Conference — ICLR 2025 Poster_

### Official Review · Reviewer_c3pF · 2024-10-28

**Soundness:** 3
**Presentation:** 3
**Contribution:** 2
**Rating:** 6
**Confidence:** 4

**Summary:**

Reviewing the classifier-free guidance (CFG), this paper found that the unconditional score in CFG can be derived by a random label vector y_hat (termed ICG) without training the unconditional model by setting the y_null label. Furthermore, the authors stimulate the guidance by introducing a perturbated time-step embedding based on the original t embedding (termed TSG). Experiments show that ICG achieves similar results to CFG, and TSG improves the FID for conditional and unconditional tasks.

**Strengths:**

1. The idea of ICG is novel and theoretically simple to implement. ICG provides a new perspective on CFG.

2. The presentation is good and the paper is easy to follow.

3. Extensive experiments show consistent effectiveness of the proposed two methods

**Weaknesses:**

1. My major concern lies in the proposed TSG:
- the authors explain TSG from its connection to Langevin dynamics, but the connection is built up on Taylor approximation where $\tilde{t} = t + \Delta{t}$, and $\Delta{t}$ should be sufficiently small. But in practice, the authors use $\Delta{t}=st^\alpha n$ where $n \sim N(0, I), s=2, \alpha=1$ and $t$ is the time-step, therefore I do think $\Delta{T}$ is sufficiently small anymore.
- I suspect the effectiveness of TSG comes from the error correction, rather than the guidance perspective, according to the results shown in Table 4.
- Will TSG work in the diffusion models that do not embed class labels into timestep (for example, UViT)?

2. please provide the NFE and samplers used to report FID throughout the paper.

3. some writing issues in section 3:
- According to eq 6 of the CFG paper [1], eq 4 in this paper looks wrong, eq 4 should be $\hat{D} = D_\theta (z_t, t, y) + w(D_\theta(z_t, t, y) - D_\theta(z_t, t, y_null))$.
- undefined notation $\beta_t$ in eq 2


[1] Ho, Jonathan, and Tim Salimans. "Classifier-free diffusion guidance." arXiv preprint arXiv:2207.12598 (2022).

**Questions:**

The authors mention that there are two methods to decide the random vector $\hat{y}$: draw from Gaussian distribution or a random class label, so that $\nabla_{z_t} log p_t (z_t | \hat{y}) = \nabla_{z_t} log p_t (z_t)$. I wonder how much would the conditional score $\nabla_{z_t} log p_t (z_t | \hat{y})$ change in practice given different samples $\hat{y}_1, \hat{y}_2, \hat{y}_3 ...$ and the same $z_t$

I would increase my rating if some of my doubts and concerns were resolved.

---

> ### Author Response · Authors · 2024-11-19
> **Official Response to Reviewer c3pF**
>
> We thank the reviewer for the positive feedback on the clarity, novelty, and strong experimental evaluation of our paper. Below, we address the concerns and questions raised, and we hope that our responses satisfactorily address the reviewer’s comments.
>
>
> ### **Theory behind TSG**
> We agree with the reviewer that the simple linear approximation often does not hold in practice. Therefore, the Langevin dynamics interpretation serves merely as an intuitive explanation for why TSG may work under idealized conditions. We believe that, in reality, TSG's influence on sampling is non-linear and involves more changes to the sampling trajectory.
>
> ### **Comment about the effectiveness of TSG**
> We partially agree with the reviewer that TSG performs error correction by filtering overly noisy and soft sampling trajectories. However, we also observed that TSG can alter the overall structure of the generated image compared to the initial unguided sampling, though not as significantly as CFG. Therefore, we conclude that the effectiveness of TSG arises from a combination of error correction and guidance.
>
> ### **TSG for diffusion models without class labels**
> We tested TSG on Stable Diffusion, which integrates conditioning into the diffusion model via cross-attention instead of adding the condition to the time-step embedding. In this case, TSG still performs effectively, as shown in Table 3. Thus, we conclude that the effectiveness of TSG is not limited by this assumption. We would be happy to evaluate the UViT model and consider including this benchmark in the final version of the paper.
>
> ### **NFE and samplers for the FID**
> We will include these details in the final version of the paper. Our experiments used multiple samplers and NFEs for each model, demonstrating that ICG/TSG are effective across various samplers and NFEs in practice.
>
> ### **Formulation of the guidance**
> There are two equivalent formulations for CFG. The difference depends on whether we consider $w=0$ or $w=1$ for the unguided case. More specifically,
> $$
> \\begin{align*}
> \hat{D}_\mathrm{CFG} &= D(z_t, t) + w(D(z_t, t, y) - D(z_t, t))\\\\
> &= wD(z_t, t, y) - (w - 1)D(z_t, t)\\\\
> &= D(z_t, t, y) + (w -1)D(z_t, t, y) - (w - 1)D(z_t, t)\\\\
> &= D(z_t, t, y) + (w -1)(D(z_t, t, y) - D(z_t, t))
> \\end{align*}
> $$
> Hence, if we define $w' = w - 1$, the CFG formulation in the paper becomes equivalent to what the reviewer mentioned in the comment.
>
> ### **Undefined $\beta_t$**
> We thank the reviewer for this suggestion. We will include a description of this parameter in the final version, i.e., it controls the influence of noise during the sampling process.
>
> ### **Question about changes in the score estimation**
> We noted that, in practice, the estimation of the score function varies slightly  depending on the random condition used as input to the network. We conducted a follow-up experiment to assess how much the sampling distribution changes when we alter the random seed for selecting $\hat{y}$ in ICG—while keeping other random factors, such as the initial noise $z_1$, exactly the same. The results are provided below.
>
> |  | FID | Precision | Recall |
> | --- | --- | --- | --- |
> | CFG | 5.56 | 0.81 | 0.66 |
> | Seed #1 | 5.55 | 0.82 | 0.65 |
> | Seed #2 | 5.54 | 0.82 | 0.65 |
> | Seed #3 | 5.55 | 0.82 | 0.65 |
>
> We observe that the final FID is relatively insensitive to this variation. Therefore, we conclude that ICG is not sensitive to this design choice. Please also refer to the variance analysis section of our response to Reviewer 8vmw for a more detailed answer to this question.

---

> > ### Comment · Reviewer_c3pF · 2024-11-23
> >
> > I appreciate the author's response and some of my concerns and doubts have been addressed.
> >
> > I want to ask one more question. When comparing CFG and ICG, do you use the same conditional model trained with class label dropping (I did not see the details from the paper)? If yes, I also wonder how would ICG perform given a pure class-conditional model.

---

> > > ### Author Response · Authors · 2024-11-23
> > > **Official Response to Reviewer c3pF**
> > >
> > > We thank the reviewer for considering our rebuttal.
> > >
> > > We evaluated ICG in both scenarios: when the model is trained with the CFG objective (e.g., DiT-XL/2) and when it is based on a purely conditional model (e.g., EDM). Our findings indicate no noticeable differences in the behavior of ICG between these scenarios.

---

> ### Comment · Reviewer_c3pF · 2024-11-23
>
> thanks for the reply, I have raised my rating. Please also add the details in the future version of the paper

---

### Official Review · Reviewer_ryvA · 2024-11-01

**Soundness:** 3
**Presentation:** 3
**Contribution:** 3
**Rating:** 6
**Confidence:** 3

**Summary:**

Classifier-free guidance (CFG) is a common method for improving the quality of samples generated from diffusion models. One limitation of CFG is the added training cost due to the requirement for both a conditional and an unconditional model. The authors propose Independent Condition Guidance (ICG), claiming that it provides the same advantages of CFG on image generation quality without additional training costs. They also propose Time-Step Guidance (TSG), a guidance algorithm based on a mixture of score estimates given both perturbed and unperturbed time-step values, which overcomes CFG’s reliance on the availability of class labels. TSG is based on a mixture of scores pertaining to a perturbed and unperturbed time-step value. The authors empirically validate their methods against CFG and un-guided baselines, demonstrating improvements in image quality over unguided sampling that are comparable in magnitude to CFG, while avoiding its limitations.

**Strengths:**

1. The paper is generally well written and easy to follow.
2. The introduction and related works sections provided a good justification for the limitations of CFG and the benefits of solutions like ICG and TSG. There is certainly a potential for this paper to provide a meaningful contribution to the field of diffusion-guidance, if my concerns about the theoretical claims and empirical validation are adequately addressed by the authors.
3. The experiments are for the most part appropriate and the ablation study is useful to understand the effect of some of the hyperparameter choices.

**Weaknesses:**

1. A central claim is that ICG provides the same benefits of CFG without the need for additional training requirements. This hinges on the validity of the theoretical claim in Section 4, which avoids the need for a potentially expensive marginalization over classes. However, the theoretical justification appears insufficient. It seems incorrect to state that sampling a class label $\hat{y}$ independently from $z_t$ at time-step $t$ implies the equivalence $\nabla_{z_t} \log p_t(\hat{y} \mid z_t) = \nabla_{z_t} \log p_t(z_t)$, since conditional independencies are a property of the diffusion model parametrized by $\theta$, irrespective of what distribution $y$ was drawn from to compute the sum in $(6)$. Given the conditional score estimate parametrized by $\theta$, $z_t$ should be dependent on $y$. The authors later allude to a procedure to "bootstrap the score", but this is not reflected in their Algorithm 1. Moreover the authors describe the correct equivalence in Appendix A, which involves marginalizing over $y$, but they don't explain how this relates to the seemingly incorrect equivalence relation (7).
2. Empirical evidence supporting TSG is limited. There are no baseline comparisons other than an unguided model. The baseline comparisons for ICG are also limited given that other methods have been published with similar motivations as ICG.
3. The choice of distribution from which $\hat{y}$ is sampled seems non-trivial, and although the authors attempted to shed some light on this in their ablation study, it is unclear how such a distribution should be chosen in general, and how sensitive are their results to the choice of distribution.
4. Several hyperparameters are introduced in the paper, but the description of the hyperparameter tuning procedure seems incomplete. Importantly, on which dataset were hyperparemeters tuned, and using what evaluation criterion?
5. Code is not made publicly available, which diminishes the paper's impact and impairs reproducibility. The provided pseudocode does not seem to entirely capture the design choices that the authors made as part of their experiments.

**Questions:**

1. Please correct or clarify the theoretical justification behind ICG.
2. Are there not other perturbation-based guidance methods, such as the ones discussed in the Related Works section (in the same spirit as SAG), that TSG could be compared to?
3. Why not include a comparison between ICG and CADS alone (Appendix B only includes ICG+CADS)? It seems like it shares many similarities with ICG. If this is the case, why is this comparison not included in the main paper?
4. The connection between TSG and Langevin dynamics is interesting, but it is not clear how the hypothesized benefits of TSG relate to, or differ from, the theoretical benefits underlying the Langevin diffusion SDE that is implicit in the SDE corresponding to the forward diffusion process.
5. Can the authors better justify the need for two hyperparameters $s$ and $\alpha$ in TSG, and how they tuned these hyperparameters (and on which dataset)?
6. The conditional and unconditional samples from the ICG method seem to have slightly more contrast and are more saturated than the CFG baseline. If the authors agree with my impression, can they hypothesize as to why this should the case?
7. While the authors may be able to justify why ICG could theoretically perform as well as CFG, I don't understand why ICG would be expected to perform better than CFG, as concluded from the plots in Figure 3. Even if the authors can justify this, it would be helpful to understand how sensitive is this result is senstive to hyperparameter choices and random seeds.
8. Figure 5: I did not understand the rationale for testing ICG against the unguided baseline as opposed to ICG against CFG on the text condition. Can the authors clarify?

Minor:
9. Can the authors revisit Section 3,4, and 5, and ensure all variables and functions are properly defined? E.g. $\epsilon$ is undefined. This would help improve the readability.
10. It seems misleading to say that Equation (1) is "equivalent to" Equation (2), and that the denoiser $D_\theta$ "approximates" the score function.
11. It would be helpful to refer to the appendix sections in relevant sections of the main text.


---Post-rebuttal: The authors have addressed the majority of my questions and concerns in their responses to my review and in their global comment. I have therefore increased my rating.

---

> ### Author Response · Authors · 2024-11-19
> **Official Response to Reviewer ryvA**
>
> We appreciate the reviewer’s positive feedback on our paper, highlighting its clear writing, strong motivation, potential significant contributions, and robust experimental results. Below, we provide responses to the questions and comments raised and remain open to further discussion.
>
> ### **Clarification on the theory of ICG**
> We thank the reviewer for highlighting this potential area of confusion regarding the theoretical justification for ICG. We would like to clarify this point and are open to further discussion. First, we acknowledge that there is a gap between the idealized theoretical scenario described in Equation 7 and the reality of practical implementation. Specifically, Equation 7 assumes access to *exact* distributions. In this ideal case, if we have a conditional distribution $p(z_t | y)$ and choose a random vector $\\hat{y}$ independent of $z_t$, we would have $p(z_t | \\hat{y}) = p(z_t)$ by definition. However, when these distributions are implicitly estimated via score-based models such as conditional denoisers, the output of the denoiser, $D_{\\theta}(z_t, t, \\hat{y}) =  \\mathbb{E}[x | z_t, \\hat{y}]$, is necessarily a function of the input arguments. We believe that it is here that the confusion in our claim is introduced. Note that while the *output* of the conditional denoiser (an estimate of the clean data $x$) is technically a function of the (random) condition, and is therefore not independent of it, our analysis in (7) relies only on the independence of the noisy *input* ($z_t$) and the condition.
>
> With real-world, limited-capacity denoisers, we concede that it will generally be the case that $D_{\\theta}(z_t, t, \\hat{y}) \\neq D_{\\theta}(z_t, t, y_{\\text{null}})$. That ICG works despite this difference offers insight into whether the *exact* unconditional component is necessary for performing CFG during sampling. Our experimental results show that by using simple formulations for each model (e.g., randomly sampling a class label), we can replicate CFG’s behavior without explicitly needing the exact unconditional score function. Even so, consistent with (15), we can interpret each random-conditional score as an unbiased estimate of the unconditional score. The results of ICG suggest that while we are not necessarily following CFG’s prescribed direction exactly at each step, both methods yield similar target distributions. However, ICG has the advantage of a simplified training process and potentially faster convergence, as shown in Figure 3, by removing the CFG label-dropout objective.
>
> The theoretical insights provided in the text and appendix are meant to offer intuition on why ICG may work in practice, without claiming this is the exact process occurring with practical denoisers. We believe we have collected considerable experimental evidence to validate the concept across various models and datasets. Nevertheless, we are grateful that the reviewer has identified an area in need of clarification, and we will modify Section 4 accordingly.
>
> ### **Empirical evidence supporting TSG**
> TSG is designed as a guidance method with minimal assumptions about the network architecture, making it applicable to all diffusion models (unlike SAG for instance, which relies on self-attention layers). Nonetheless, we conducted experiments comparing TSG with SAG and PAG using Stable Diffusion. The results provided below show that TSG outperforms both methods.
>
> | Guidance method | FID | Precision | Recall |
> | --- | --- | --- | --- |
> | no guidance | 69.38 | 0.42 | 0.49 |
> | SAG [1] | 59.34 | 0.48 | 0.53 |
> | PAG [2] | 62.85 | 0.42 | 0.51 |
> | TSG | **56.65** | **0.54** | **0.54** |
>
> [1] Hong, Susung, Gyuseong Lee, Wooseok Jang and Seung Wook Kim. “Improving Sample Quality of Diffusion Models Using Self-Attention Guidance.” *2023 IEEE/CVF International Conference on Computer Vision (ICCV)* (2022): 7428-7437.
>
> [2] Ahn, Donghoon, Hyoungwon Cho, Jaewon Min, Wooseok Jang, Jungwoo Kim, SeonHwa Kim, Hyun Hee Park, Kyong Hwan Jin and Seungryong Kim. “Self-Rectifying Diffusion Sampling with Perturbed-Attention Guidance.” *ArXiv* abs/2403.17377 (2024): n. pag.
>
> ### **Baselines for ICG**
> ICG aims at replicating the behavior of CFG, which is why CFG is used as the ground truth for comparison. For this reason, the method is compared exclusively with CFG. If the reviewer has a specific benchmark they believe ICG should be evaluated against, we would be happy to include it in the final version.
>
> ### **Distribution for $\hat{y}$**
> Our goal is to identify independent conditions that can be fed into the model. For class labels and text tokens, this involves using random class labels or random text tokens, which can be generated through random integer sampling. However, we believe this is not the only approach for selecting random conditions, as demonstrated in our ablation study using a Gaussian noise vector. Exploring the optimal distribution for $\hat{y}$ represents an interesting direction for future work.

---

> ### Author Response · Authors · 2024-11-19
> **Official Response to Reviewer ryvA (part 2)**
>
> ### **Hyperparameters used in the paper**
> Since ICG/TSG are inference methods, the hyperparameters were selected by generating a small set of images and using visual inspection. We did not conduct tuning on a specific dataset or perform a grid search. It is quite possible that better hyperparameters exist for each model.
>
> ### **Code availability**
> Unfortunately, due to internal copyright policies, we cannot share the full code for the paper. However, beyond the implementation details already provided, we will revisit Section E in the final version of the paper to ensure reproducibility.
>
> ### **Comparing ICG with CADS**
> CADS is a method to enhance the diversity of CFG, serving as an addition to regular CFG, whereas ICG is intended to replicate the base CFG behavior. Therefore, a direct comparison between CADS and ICG is not possible, as CADS is not designed to function independently without CFG. However, we can include the comparison between CFG + CADS and ICG + CADS as shown below, where we note that ICG + CADS slightly outperform CFG + CADS in terms of FID.
> | method | FID |
> | --- | --- |
> | CFG + CADS | 9.47 |
> | ICG + CADS | **8.83** |
>
> ### **Connection between TSG and Langevin dynamics**
> This part aims to show that if we use an ODE as the diffusion solver and use TSG-guided output instead of the output of the diffusion model, under the idealized linearity assumption, the ODE update step will be transformed to an SDE. Since SDE-steps have been seen to improve the sampling quality due to error correction, we concluded that TSG-steps are to some extent following the same logic (at least under this idealized assumption). However, we agree that in practice, TSG is doing non-linear corrections to the trajectories, and its effect is closer to CFG rather than changing the ODE to SDE. The DiT model uses an SDE sampler (DDPM), and we still observed significant improvements by using TSG on top of this sampler.
>
> ### **Question about $s$ and $\alpha$ in TSG**
> $s$ is intended to scale the noise such that its variance aligns with the scale of the time-step embedding. We observed that deviating too far from the time-step embedding disrupts the diffusion model's ability to perform denoising correctly. Meanwhile, $\alpha$ is designed to progressively reduce the perturbation to zero as denoising advances, limiting guidance in the final steps. The parameters were selected through visual inspection and were not tuned on any specific dataset.
>
> ### **Contrast in ICG samples**
> Since ICG updates differ slightly from CFG at each step, the effect of the guidance weight also varies. The figures were generated using the same guidance scale for both ICG and CFG. By slightly lowering the guidance scale for ICG, we can achieve the same contrast and saturation levels across both images, as ICG updates are observed to be slightly stronger than those of CFG.
>
> ### **Explanation for Figure 3**
> We do not claim that ICG is better than CFG. In Figure 3, our goal is to demonstrate that the training budget allocated to the CFG objective could be better utilized to train the conditional model itself. The figure shows that applying ICG to a purely conditional model results in better FID at each checkpoint compared to a model that dedicates 10% of its training to the CFG label-dropout objective. This suggests that focusing all training resources on the conditional model and using ICG at inference can achieve CFG-like benefits without any training requirements. We observed no sensitivity of these results to the hyperparameters, with the performance improvement coming from the ICG model being effectively trained more due to not including the CFG objective.
>
> ### **Question about Figure 5**
> For Figure 5, we evaluated ICG on image-conditioned models, specifically for pose-to-image and depth-to-image generation. We chose ControlNet as a good open-source model for these tasks. Our intention is not to suggest ignoring text conditions for this specific model, but rather to demonstrate that the effect of ICG is not limited to class labels or text conditions. In this context, ControlNet without text serves as an example of a pure image-to-image model, accommodating scenarios where such models may not have access to text conditions.
>
> ### **Minor comments**
> We will review the manuscript thoroughly and incorporate the necessary revisions in the final version to address the reviewer's comments.

---

> > ### Comment · Reviewer_ryvA · 2024-11-23
> >
> > I appreciate the author’s detailed response. The majority of my questions and concerns were satisfactorily addressed, and will lead me to increase my score. However, I may increase my score further if the authors can address the following questions/suggestions, which I believe could improve the clarity and impact of the paper:
> >
> > 1. Hyperparameter tuning procedure: Although I am aware of the resources required to tune each model, and am not necessarily suggesting that you provide a more rigorous procedure that ensures equitable comparison of all models, I believe your paper should present a version of your response to my question regarding how you tuned your hyperparameters. This would help readers understand the possible impact of your hyperparameter tuning procedure on the reproducibility of your results.
> >
> > 2. “if we have a conditional distribution $p(z_t | y)$ and choose a random vector $\hat{y}$ independent of $z_t$, we would have $p(z_t | \hat{y}) = p(z_t)$ by definition.” Although I intuitively see what you mean, I still feel like the way it is explained might lead to confusion. You have a distribution $p(z_t | y)$, and we know that in $p$, $z_t$ and $y$ are dependent (as you seemed to agree in your response to my comment). Independence is a property of the distribution $p$, not a property of the value assigned to the random variable $y$. If you assign $y=\hat{y}$, then by definition $p(z_t | y=\hat{y}) \neq p(z_t)$, irrespective of the distribution from which $\hat{y}$ may have been sampled. What you said in the sentence I quoted above is different from saying that you can design an estimator of $p(z_t)$ that is based on samples $\hat{y}$ drawn from some other distribution that does not depend on the value of $z_t$, and making a claim that it is an unbiased estimator (which involves an expectation). Even though a single ICG estimate of the unconditional score may not be equal to the true unconditional score, you can derive conditions under which your proposed estimator is unbiased. The expected error that you derived in a toy setting as part of your response to reviewer 8vmw (see: https://openreview.net/forum?id=b3CzCCCILJ&noteId=b89mS0TzH2) is in line with what I am suggesting. Unless you disagree with my interpretation, I would therefore recommend that you include in your revision a derivation (or refer to it in an Appendix) of basic properties (bias/variance) of your estimator, even in such a toy setting. This could help the reader better understand the theoretical justification behind ICG, compared to simply stating Eq (7). It would also allow the reader to gain insight into the effect of the choice of distribution from which $\hat{y}$ is sampled on the bias/variance of the estimator.
> >
> > 3. “Even so, consistent with (15), we can interpret each random-conditional score as an unbiased estimate of the unconditional score.” Is it correct to state that the ICG-estimator is unbiased?  It seems as though you have shown in your response to reviewer 8vmw (https://openreview.net/forum?id=b3CzCCCILJ&noteId=b89mS0TzH2) that the expected error is in general non-zero and depends on the distribution from which $\hat{y}$ is sampled?

---

> ### Author Response · Authors · 2024-11-23
> **Official Response to Reviewer ryvA**
>
> We thank the reviewer for providing valuable and constructive additional feedback on the paper. Below, we address each individual comment and question in detail. Additionally, we invite the reviewer to refer to the global comment for further information.
>
> ### **Comment on hyperparameter selection**
> We are happy to include our rule of thumb for selecting hyperparameters in the final version of the paper. The noise scale $s$ was chosen by observing the standard deviation of the time step embedding $t_{emb}$ and selecting values around it. For instance, choosing $s = 2 \times \text{std}(t_{emb})$. The parameter $\alpha$ controls how the perturbation should change over the denoising process. We observed that values within the range $[0.25, 1.5]$ generally work well across different settings. Higher values of $\alpha$ result in less perturbation, keeping the results closer to the base model.
>
> As shown in the ablation studies, excessive perturbation may reduce FID, whereas insufficient perturbation often yields only minor improvements over the baseline. For each model, the exact hyperparameter values were selected by generating a small batch of images under various parameter settings and performing a quick visual inspection.
>
> ### **Comment on ICG theory**
> We are grateful to the reviewer for pressing us on this point, as further analysis has revealed what we believe to be a much more satisfying theoretical explanation for ICG’s success in providing the benefits of CFG without an unconditional model. Please see our general response above under “Bounding the Error on the Unconditional Score Approximation.”
>
> The reviewer correctly points out that independence is a property of the distribution $p$ and not the choice of condition. It might be helpful to mention our original intuition, which takes some inspiration from permutation testing for independence. Ignoring the noisy $z$ variables for the moment, consider the distribution $p(x,y)$ representing the original data and their corresponding conditions. This is represented implicitly in the paired data $\{(x_i,y_i)\}$. In permutation testing, one shuffles the data to form new pairs $\{(x_i,y_j)\}$ for $i \neq j$ in order to break the dependencies. Such permutations create a new distribution $\tilde{p}(x,y) = p(x)p(y)$ in which $x$ and $y$ are independent by design. This motivated our approach to ICG, and when we empirically found that it works across a variety of models and datasets, the original intuition continued to make sense. However, the actual situation turns out to be more nuanced, which we address in the above-referenced global comment.
>
> ### **Comment on ICG bias**
> If one samples a condition from $p(y|z)$, then the resulting conditional score is indeed an unbiased estimate of the unconditional score. A second condition in which the random-conditional score is an unbiased estimate of the score is when $z$ and $y$ are independent. But as we acknowledge in the point directly above and address in our global comment on the score estimate, neither condition exactly applies in our case. As a result, our referring to the random-conditional score in our original rebuttal remarks as an *unbiased* estimate of the unconditional score is true only in idealized circumstances that do not necessarily apply in practice. We thank the reviewer for challenging this point. Please also see our global comment above.

---

> > ### Comment · Reviewer_ryvA · 2024-11-25
> >
> > Thank you for your responses here and in the global comment. I have increased my rating.

---

### Official Review · Reviewer_8vmw · 2024-11-03

**Soundness:** 2
**Presentation:** 3
**Contribution:** 2
**Rating:** 6
**Confidence:** 4

**Summary:**

This paper proposes two methods, independent condition guidance (ICG) and time-step guidance (TSG), for conditional and unconditional sampling from a diffusion model, which are summarized below.

**ICG**

They propose ICG in place of classifier-free guidance (CFG) such that instead of learning two separate models or having two different tasks, one conditional and one conditional, they learn a single conditional model with only a conditional task without making use of null tokens. They do based on the insight that:

1. A conditional model can be turned into an unconditional model by noting that:
    1. the marginal distribution $p(x_t)$ is equal to the expectation $E_{y \sim p(y)}[p(x|y)]$ which can be estimated using sampling a single independent sample $y \sim p_{data}(y)$, which implies that $s_\theta(x_t) \approx s_\theta(x_t, y)$.

They propose ICG to “streamline” the training of conditional model and improve efficiency (line 534) and as an alternative to CFG, which requires either separate models or a single model with a null token, representing unconditional generation. With ICG, the authors show that training a conditional model is sufficient and the CFG update can be replaced by:

$D_{ICG} = D(z_t, t, \widehat{y}) + w_{ICG}(D(z_t, t, y) - D(z_t, t, \widehat{y}))$

where $\widehat{y}$ is a Gaussian vector or a class sampled independent of $y$ at each time-step $t$.

**TSG**

TSG is based on the insight that score model architectures add the class-conditioning embedding into the time-step embedding. Therefore, similar to ICG, they perturb the time-step $t$ by $\Delta t \sim N(0, \sigma)$ TSG then uses a linear combination of $D_\theta(x_t, t)$ and $D_\theta(x_t, t + \Delta t)$ as the “score” network update. The justification of TSG is based on a connection to Langevin dynamics.

In practice, they do not perturb the time, but similar to ICG they perturb the time-step embedding vector by $s t^{\alpha} n$, where $s, t, \alpha$ selected to match the mean and scale of the “real” time-step embedding vectors and $n \sim N(0, 1)$, and they propose the following update instead of CFG:

$D_{TSG} = D(z_t, \tilde{t}, y) + w_{TSG}(D(z_t, t, y) - D(z_t, \tilde{t}, y))$ where $\tilde{t}$ is the perturbed embedding.

**Strengths:**

The paper is well-written and the authors perform extensive experiments and provide algorithmic and experimental details to reproduce their experiments.

**Experiments**

The experiments show that ICG and CFG perform identically and ICG “simulates the behavior of CFG across several conditional models” (line 249). They also run a number of ablations, for instance, they compare using an independent Gaussian vector versus sampling $y \sim p(y)$ in ICG, providing stronger experimental evidence.

**Weaknesses:**

The main question left un-answered by this work is whether a marginal model, either trained separately or with a null token, is necessary for a sampling method like CFG:
1. An analysis of the variance of the estimate $s_\theta(x_t \mid y)$ is missing from the paper.
2. An analysis of the impact of the variance on sampling is also missing from the paper
3. Is there a trade-off between the slower convergence of training with a null token with estimation of the marginal model?

See the questions section.

The authors could improve the paper by providing a proof (or an empirical analysis) for the sampling procedure for a toy distribution where the $p_{data} = 0.5 N(-a, 1) + 0.5 N(a, 1)$. For instance, with a mixture of Gaussians as the data distribution, the marginal and conditional score functions are computable in closed form and a toy experiment, where various trade-offs can be examined, could make the paper stronger.

TSG
For connecting TSG to Langevin dynamics, the authors show that eq 9 resembles a Langevin dynamics step. However, by using the perturbed TSG denoising model $\widehat{D}$  to define the score $\nabla_{z_t} \log \widehat{p}(z_t)$, the connection seems circular. Could the authors clarify what the connection to Langevin dynamics is and the relevance of that connection?

**Questions:**

The authors present an estimate of the marginal model $p(x_t)$, where instead of marginalizing over all possible $y$ by learning a score model with a null token, they use a Monte Carlo estimate. However, the variance of the estimate of the marginal score and it’s impact on sampling has not been addressed in the paper.

1. Supports of $p(x_t | y=1)$ and $p(x_t | y=0)$ are different: In case of two classes and when $p(x_t | \widehat{y})$ is low, in such a case the ICG term is the score of $p(x_t | \widehat{y})^{1 - w_{ICG}} p(x_t | y)^{w_{ICG}}$, if $p(x_t | \widehat{y})$ is low then its score’s magnitude is high and would dominate the ICG update, potentially pushing in the direction of $p(x_0 | \widehat{y})$
    1. will more steps be required for ICG sampling compared to CFG?
    2. What will happen when the support for $x_t$ all $t \in [0, T]$ is disjoint for the two classes, such that $p(x_t | \widehat{y}) = 0$ when $\widehat{y}$ is the wrong class? For instance, if $p_{data} = 0.5  N(-5, 1) + 0.5 N(5, 1)$, and the model prior is a centered Gaussian.
2. If the classes are imbalanced then sampling from the majority class can bias the sampling process when the user wishes to sample from the minority class. For instance, if $p(y = 0 | x_0) = 0.99$ and $p(y = 1 | x_0)= 0.01$,  and ICG samples $\widehat{y}$ with an equal probability. The authors should consider modifying the algorithm to sample from $p(y)$, when available, and not a uniform distribution.

Some simple theoretical guidance with/or toy examples should suffice for an explanation. Potentially with $p(x_0)$  as a mixture of Gaussians in 2d and the authors could vary the guidance scale while sampling.

1. For text to image sampling with ICG, do the authors recommend sampling random text prompts to estimate the marginal model?
2. For the text conditional experiments in Table 3, can the authors also include CFG performance as a baseline?
3. For the ControlNet experiment, I believe the authors describe training with an empty string for 50% of their text prompts. See section 3.3 in the ControlNet paper. Do the authors use a random text and/or image prompt for the ControlNet model to define a marginal model?
4. In line 053, the authors claim that there is no clear extension of CFG to unconditional generation:
    1. However, CFG requires training a marginal model, which can be sampled easily. Moreover, one could randomly sample a label y to do generation as well.
    2. Can the authors provide details about unconditional sampling with ICG? The update defined in algorithm 1, requires a condition $y$ as input. is it just randomly sampling a label and then running ICG?

    $D_{ICG} = D(z_t, t, \widehat{y}) + w_{ICG}(D(z_t, t, y) - D(z_t, t, \widehat{y}))$

---

> ### Author Response · Authors · 2024-11-19
> **Official Response to Reviewer 8vmw**
>
> We thank the reviewer for finding the paper well-written, with extensive experiments and good ablations. We hope that the comments below satisfactorily address the reviewer’s questions and concerns.
>
> We would like to begin with the main question the reviewer poses, namely whether the unconditional score is necessary for obtaining the quality-boosting benefits of CFG. We believe that our work demonstrates that the unconditional score is *not* necessary. We have demonstrated that independent condition guidance (ICG) is sufficient for achieving CFG-like benefits across various models and datasets without ever training an unconditional model in any form.
>
> We have also provided detailed responses to each question and comment below, and we welcome further discussion.
>
> ### **Variance analysis**
>
> We thank the reviewer for raising this question. We address it in two parts:
>
> 1. How much each intermediate sampling step $z_t$ differs between ICG and CFG.
> 2. How much the final sampled distribution is affected by these differences.
>
> For the first part, we define the relative error for each trajectory as
> $$e_t = \\frac{\\Vert z_t^{\\mathrm{ICG}} - z_t^{\\mathrm{CFG}}\\Vert}{\\Vert z_t^{\\mathrm{CFG}} \\Vert}.$$
>
> We computed $e_t$ for different random class labels as our choice of $\\hat{y}$ in ICG using the DiT model and observed that it is on average below 0.0025 (or 0.25%) across all time steps.
>
> For the second question, we measured how different random seeds for choosing $\\hat{y}$ in ICG affect the final distribution, using CFG as the baseline and 10,000 generated samples. The results are as follows:
>
> |  | FID | Precision | Recall |
> | --- | :---: | :---: | :---: |
> | CFG | 5.56 | 0.81 | 0.66 |
> | ICG Seed #1 | 5.55 | 0.82 | 0.65 |
> | ICG Seed #2 | 5.54 | 0.82 | 0.65 |
> | ICG Seed #3 | 5.55 | 0.82 | 0.65 |
>
> The results above indicate that the ICG trajectory is very close to the CFG sampling trajectory, and that slight variations between CFG and ICG at each step do not lead to any noticeable change in the target distribution.
>
> ### **ICG analysis based on a mixture of Gaussians**
> If we assume a Gaussian distribution for each class, with
> $$
> p(x \\mid y=0) = \\mathcal{N}(\\mu_0, I) \\quad \\text{and} \\quad p(x \\mid y=1) = \\mathcal{N}(\\mu_1, I),
> $$
> then after the forward diffusion process, the conditional distributions become:
> $$
> p(z_t \\mid y=0) = \\mathcal{N}(\\mu_0, (1 + \\sigma_t^2)I) \\quad \\text{and} \\quad p(z_t \\mid y=1) = \\mathcal{N}(\\mu_1, (1 + \\sigma_t^2)I).
> $$
> and the unconditional probability is equal to (assuming uniform prior for each class)
> $$
> p(z_t) = 0.5p(z_t \\mid y=0) + 0.5p(z_t \\mid y=1).
> $$
> The conditional score functions in this case are given by:
> $$
> s(z_t \\mid y=0) = -\\frac{z_t - \\mu_0}{1 + \\sigma_t^2}, \\quad \\text{and} \\quad
> s(z_t \\mid y=1) = -\\frac{z_t - \\mu_1}{1 + \\sigma_t^2}.
> $$
> The unconditional score can be expressed as:
> $$
> s(z_t) = \\frac{0.5p(z_t \\mid y=0)}{p(z_t)} \\left(-\\frac{z_t - \\mu_0}{1 + \\sigma_t^2}\\right) + \\frac{0.5p(z_t \\mid y=1)}{p(z_t)} \\left(-\\frac{z_t - \\mu_1}{1 + \\sigma_t^2}\\right).
> $$
> In this case, the CFG update direction will simplify to
> $$
> s(z_t \\mid y=0) - s(z_t) = \\frac{p(z_t \\mid y=1)}{p(z_t \\mid y=0) + p(z_t \\mid y=1)} \\left(\\frac{\\mu_0 - \\mu_1}{1 + \\sigma_t^2}\\right).
> $$
> Next, let’s compute the ICG update. For this analysis, we assume that the random label $\\hat{y}$ is drawn from a Bernoulli distribution, i.e., $p(\\hat{y} = 0) = p_0$. In expectation, the ICG estimate of the unconditional score is equal to
> $$
> \\overline{s(z_t)} = p_0 s(z_t \\mid y=0) + (1 - p_0) s(z_t \\mid y=1).
> $$
> This gives us
> $$
> \\begin{align*}
> s(z_t \\mid y=0) - \\overline{s(z_t)} &= (1 - p_0) (s(z_t \\mid y=0) - s(z_t \\mid y=1)) \\\\
> &= (1-p_0) \\left( -\\frac{z_t - \\mu_0}{1 + \\sigma_t^2} + \\frac{z_t - \\mu_1}{1 + \\sigma_t^2} \\right) \\\\
> &= (1-p_0) \\left(\\frac{\\mu_0 - \\mu_1}{1 + \\sigma_t^2}\\right)
> \\end{align*}
> $$
> This analysis shows that in theory, there is a certain choice for $p(\\hat{y} = 0)$ such that the update direction of ICG and CFG exactly match in expectation. In every case, however, the expected update is always in the *same direction* as the CFG update rule.
>
> In real-world problems, finding the categorical distribution that exactly matches the magnitude of the CFG direction is not straightforward. However, we experimentally showed that using a simple uniform distribution for $p(\\hat{y})$ is good enough to simulate the behavior of CFG. We will include a more detailed and generalized version of this analysis in the final version of the paper.
>
> ### **Question about the number of steps for ICG**
> ICG requires the same number of steps as CFG. For Table 1 and 2, we tested the models for several NFEs and ICG results were similar or better than CFG across all of our experiments.

---

> > ### Author Response · Authors · 2024-11-19
> > **Official Response to Reviewer 8vmw (part 2)**
> >
> > ### **Questions about the support of the conditional distributions**
> > Please note that the guidance term always pushes the sample away from $p(z_t|\\hat{y})$. Hence, even for areas where this probability is low, we do not go toward this function but rather away from this distribution, as we have $w_{\\mathrm{CFG}} > 1$. Additionally, since the input $z_t$ is noisy, the distributions cannot be disjoint for all time steps. For instance, we always have $p(z_1 | y) = \\mathcal{N}(0, \\sigma_{\\mathrm{max}}{I})$ for all $y$, which is nonzero over the whole space. We believe that the distributions can become disjoint toward the end of sampling, but it has been observed that such steps do not contribute toward the guidance outputs [1].
> >
> > [1] Kynkäänniemi, Tuomas, Miika Aittala, Tero Karras, Samuli Laine, Timo Aila, and Jaakko Lehtinen. "Applying guidance in a limited interval improves sample and distribution quality in diffusion models." arXiv preprint arXiv:2404.07724 (2024).
> >
> > ### **Question about imbalanced classes**
> > We thank the reviewer for raising this interesting question. In the paper, we explored straightforward methods for obtaining $\hat{y}$ for ICG. We agree that better choices may exist depending on the specific problem, as shown in the Gaussian mixture analysis above. We will gladly mention this discussion in the final version of the paper.
> >
> > ### **Question about the convergence trade-off**
> > We believe that ICG involves no trade-off in this regard. Empirically, we demonstrated that by simply choosing a random condition at each time step, we can replicate the benefits of CFG without requiring any training for the unconditional model. Furthermore, we showed that the training budget allocated to the CFG objective can be more effectively used to train the conditional model. These findings suggest that the exact unconditional distribution may not be necessary for CFG, and ICG offers a sufficient approximation without compromising quality.
> >
> > ### **Clarification on the Relation Between TSG and Langevin Dynamics**
> > The methodology for this analysis closely mirrors that of CFG. Consider an ODE solver designed to follow the conditional score function $s(z_t \\mid y)$. For CFG, this score function is replaced with the CFG-guided output $\\hat{s}\_{\\mathrm{CFG}}(z_t \\mid y)$. In our Langevin dynamics analysis, we applied the same approach, substituting the conditional score with the corresponding score function derived from the TSG update rule, $\\hat{s}\_{\\mathrm{TSG}}(z_t \\mid y)$. We then demonstrated that, under certain approximations, this substitution introduces a Langevin dynamics step into the ODE solver. We are happy to provide further discussion if additional clarification is needed.
> >
> > ### **Random condition for the text-to-image models**
> > In our early experiments, we tested both random text tokens (based on the CLIP model for Stable Diffusion) and random text generated by combining random characters into sentences of variable length. Both approaches yielded similar performance in our tests, demonstrating ICG's flexibility w.r.t. the random condition selection.
> >
> > ### **CFG baseline in Table 3**
> > The reason for excluding CFG from Table 3 is that TSG is intended to complement CFG, rather than replace it when a conditional model is available. We aimed to demonstrate that TSG applies to both conditional and unconditional models. In Table 4, we showed that TSG can be combined with CFG to enhance generation quality. We are happy to include text-to-image results based on Stable Diffusion in Table 4, illustrating that combining CFG and TSG outperforms each method used in isolation.
> >
> >
> > ### **Question about ControlNet**
> > We used ControlNet without a text prompt as an open-source model for image-to-image tasks in this experiment to demonstrate that ICG can also be applied to image-conditioned models. Thus, we used random pose/depth maps (obtained via Gaussian noise) to estimate the marginal model for the image condition input in this scenario.
> >
> > ### **Using CFG for unconditional models**
> > CFG assumes the availability of both a conditional model $p(z_t | y)$ and an unconditional model $p(z_t)$. However, when the dataset lacks labels (e.g., FFHQ), it is not possible to train a conditional model $p(z_t | y)$, making CFG inapplicable in such scenarios where only $p(z_t)$ is available. While it is possible to perform unconditional generation on a class-conditioned model by randomly selecting a class label (assuming balanced labels), this approach is not universally applicable, as a conditional model is not always available.

---

> > > ### Comment · Reviewer_8vmw · 2024-11-22
> > >
> > > Thanks for such a thorough response. I believe most of my concerns have been answered. I list the remaining concerns below. I would be happy to improve my scores if the authors can answer these questions.
> > >
> > > Can the authors clarify the benefit, and a comparison, of their approach over that of [Karras et al 2024], which also replaces the marginal model in CFG with a conditional model. The FID scores for their approach is significantly better and their method also does not require a marginal model.
> > >
> > > In table 2, the report FID score for EDM2-XS is higher than the number reported by the EDM2 paper, see figure 1. Is there any explanation for this?
> > >
> > > For the variance analysis, it would be better to do for the GMM example as well as if the authors could provide details regarding what exactly they ran on ImageNet, for instance:
> > >
> > > 1. how many samples were used
> > > 2. how many classes were sampled
> > > 3. rather than the sampling step, i was asking about the variance of the independent condition score approximation to the marginal score, that is a comparison between $s_\theta(x_t, t, \hat{y})$ and $s_\theta(x_t, t)$.
> > >
> > > While it is true that in expectation the ICG gradient is equal to the marginal, my question was about the impact of such adversarial data distributions on the ICG sampling scheme, where the authors do not use the expected value of the independent condition score $s_\theta(x_t, t, \hat{y})$.
> > >
> > > For the support, it is true that towards the end of sampling the inference process the supports of p(x_t|y=1) and p(x_t | y=0) match since the process mixes, while generating samples the supports will get disjoint and sampling the opposing class would in effect amount to a step in the opposing direction. Would it be possible for the authors to add a 2d experiment with 0.1 N(-5, 0.1) + 0.9 N(+5, 0.1) as their data distribution and run an analysis of ICG with various hyper-parameter choices such as the $p_0$ distribution, number of time-steps.
> > >
> > > [Karras et al 2024] Guiding a Diffusion Model with a Bad Version of Itself

---

> ### Author Response · Authors · 2024-11-22
> **Official Response to Reviewer 8vmw**
>
> We appreciate the reviewer’s time in considering our rebuttal and engaging in the discussion. Please find our responses below.
>
> ### **Comparison with Autoguidance**
> Autoguidance [1] is a concurrent work that leverages a second, intentionally inferior diffusion model to guide sampling in order to address some of CFG’s drawbacks, such as low sample diversity. By contrast, ICG simplifies the application of CFG by eliminating its dependency on an unconditional model but still shows the same quality-boosting benefits. While autoguidance, like our method, does not require an unconditional model, it does require training a smaller version of the conditional model for part of the original training process, making it unsuitable for direct application to pre-trained models, unlike ICG. For text-to-image models, the autoguidance paper suggests that the benefits of CFG and autoguidance are complementary, leading the authors to combine both methods for potentially improved text-image alignment. In this context, ICG could serve as a replacement for the CFG component in such combinations. Also, as we show in the paper, ICG is compatible with diversity-boosting techniques such as CADS.
>
> ### **FID in Table 2**
> The FID reported in the EDM2 paper was achieved through an extensive grid search across various combinations of guidance weights and EMA profiles for both conditional and unconditional models. Due to computational constraints, performing an equivalent grid search for ICG was not feasible. Instead, we compared the models using a different guidance scale (not optimized for FID) to ensure a fair evaluation. It is important to note, however, that both models achieve identical FD_Dino scores, which have been shown to correlate more closely with human judgment. We conclude that it is possible to identify specific parameters for ICG (in terms of EMA profiles + guidance scales) that replicate the FID reported in the original paper.
>
> ### **Variance analysis and the support of the distributions**
> The analysis was performed using a batch size of 32, with 10 different random seeds for choosing the random condition in ICG. We can perform the same analysis and compute the error for the score estimation defined as
> $$
> e_t = \\frac{\\Vert s(z_t, t, \\hat{y}) - s(z_t, t)\\Vert}{\\Vert s(z_t, t) \\Vert}.
> $$
> In this case, the same computation gives us around 0.049 error for $e_t$ averaged over time steps. Note that the primary aspects of interest when sampling from diffusion models are the final modeled distribution and the sampling trajectories rather than the precise intermediate model predictions. Our empirical results show that slight variations in model predictions can still lead to the same final target distribution, as evidenced by almost the same FID values achieved by ICG and CFG.
>
>
> We can also follow the error analysis using the GMM model. In this case, the error term for each prediction is given by:
> $$
> s(z_t) - s(z_t, \\hat{y}=0) =   \\frac{p(z_t \\mid y=1)}{p(z_t \\mid y=0) + p(z_t \\mid y=1)} \\left(\\frac{\\mu_1 - \\mu_0}{1 + \\sigma_t^2}\\right),
> $$
> $$
> s(z_t) - s(z_t, \\hat{y}=1) =    \\frac{p(z_t \\mid y=0)}{p(z_t \\mid y=0) + p(z_t \\mid y=1)} \\left(\\frac{\\mu_0 - \\mu_1}{1 + \\sigma_t^2}\\right).
> $$
> Accordingly, the average error (in terms of norm) w.r.t. $p(\\hat{y})$ will be equal to:
> $$
> \\begin{align*}
> \\overline{e_t} &= p(\\hat{y}=0)\\Vert s(z_t) - s(z_t, \\hat{y}=0)\\Vert + p(\\hat{y}=1)\\Vert s(z_t) - s(z_t, \\hat{y}=1)\\Vert \\\\
> &= p_0 \\frac{p(z_t \\mid y=1)}{p(z_t \\mid y=0) + p(z_t \\mid y=1)} \\left(\\frac{\\Vert \\mu_0 - \\mu_1 \\Vert}{1 + \\sigma_t^2}\\right) + (1 - p_0) \\frac{p(z_t \\mid y=0)}{p(z_t \\mid y=0) + p(z_t \\mid y=1)} \\left(\\frac{\\Vert \\mu_0 - \\mu_1 \\Vert}{1 + \\sigma_t^2}\\right).
> \\end{align*}
> $$
> Recall that to match the expectation, one should choose
> $$
> p_0 = \\frac{p(z_t \\mid y=0)}{p(z_t \\mid y=0) + p(z_t \\mid y=1)},
> $$
> which gives us
> $$
> \\overline{e_t} = 2\\frac{p(z_t \\mid y=0)p(z_t \\mid y=1)}{[p(z_t \\mid y=0) + p(z_t \\mid y=1)]^2} \\left(\\frac{\\Vert \\mu_0 - \\mu_1 \\Vert}{1 + \\sigma_t^2}\\right).
> $$
> In this idealized formulation, if the distributions are relatively disjoint, the expected error term vanishes, as it depends on the product $p(z_t \\mid y=0) p(z_t \\mid y=1)$. We hope this explanation clarifies the reviewers' concerns regarding disjoint class distributions. Given the time constraints for the rebuttal, we would be happy to include a more detailed numerical error analysis (using both images and 2D distributions) in the final version of the paper.

---

> > ### Comment · Reviewer_8vmw · 2024-11-23
> >
> > The analysis presented by the authors shows that when we are sampling $p(z_0 |y)$, then if $z_t \sim p(z_t | y=1)$  implies that $p_0(t) = \frac{p(z_t | y=0)}{p(z_t | y = 0) + p(z_t | y = 1)}$, with disjoint supports, is approximately zero. So in the idealized instance, ICG update would reduce to sampling the conditional score update.
> >
> > So is the following a correct summarization:
> > 1. the ICG update is an unbiased estimate of CFG when the distribution $p(y)$ is known in advance
> > 2. if the distribution $p(y)$ is not known in advance in that case, the ICG update is biased?
> >
> > It would be useful to see some analysis of this in a few settings.
> >
> > For a fairer comparison, it would be better to report the best possible numbers of CFG in table 2. However, can the authors make it clearer why they believe DINO FID for both guidance methods is equivalent when the authors did not choose the best CFG setting, albeit for FID, provided by the authors of EDM2.

---

> ### Author Response · Authors · 2024-11-23
> **Official Response to Reviewer 8vmw**
>
> We thank the reviewer for taking the time to consider our response. Below, we provide a detailed reply to the specific comments raised, while also referring to the global comment above for additional context.
>
> ### **Comment about bias in the score estimation**
> The intuition of the reviewer is correct that ICG is only unbiased under some idealized circumstances. We have revisited our analysis of the unconditional score estimator and have what we believe is a more satisfying theoretical perspective on it in our global remark under “Bounding the Error on the Unconditional Score Approximation.” We thank the reviewer for pressing this point, and we will include this bias analysis in the final version of the paper.
>
> ### **Comment about FD_DINOv2**
> We will revise the CFG FID results for EDM2 to ensure a fair comparison. The EDM2 authors’ hyperparameter search had results that are both method (EDM, CFG) and metric (FID, FD_DINOv2) specific. The choices that optimize one metric will not necessarily optimize the other, nor will they transfer across methods. The default settings in the EDM2 codebase allow us to reproduce the best FD_DINOv2 results reported in the original paper using CFG, which we are able to slightly improve upon using ICG with no separate hyperparameter search. Thus, the non-optimized setting we mentioned in our previous response only applies to FID.

---

### Author Response · Authors · 2024-11-23
**Official Response to Reviewers**

We thank all reviewers for their time and for providing valuable feedback to enhance the argumentation surrounding ICG. In response to the reviewers’ comments and suggestions, we offer the following detailed explanation regarding the bias of ICG. Based on this constructive exchange, we will modify Section 4 (and the appendix) in the final version of the paper to incorporate the points discussed during the rebuttal.

### **Bounding the Error on the Unconditional Score Approximation**
It can be shown (see Equation 15 in our paper) that
$$
\\mathbb{E}\_{p(y|z)} \\left[ \\nabla_z \\log p(z|y) \\right] = \\nabla_z \\log p(z).
$$
When the expectation is instead taken over the distribution $q(y)$, we have
$$
\\mathbb{E}\_{q(y)} \\left[ \\nabla_z \\log p(z|y) \\right] = \\nabla_z \\log p(z) - \\nabla_z \\mathcal{D}\_{\\text{KL}}(q(y)\\|p(y|z)).
$$

*Proof.* We can write $\\nabla_z \\log p(z|y) = \\nabla_z \\log p(z,y) - \\nabla_z \\log p(y) = \\nabla_z \\log p(z,y)$, which in turn can be written as $\\nabla_z \\log p(z,y) = \\nabla_z \\log p(z) + \\nabla_z \\log p(y|z)$. Taking the expectation over $q(y)$, we have
$$
\\mathbb{E}\_{q(y)} \\left[ \\nabla_z \\log p(z,y) \\right] = \\nabla_z \\log p(z) + \\mathbb{E}\_{q(y)} \\left[ \\nabla_z \\log p(y|z) \\right] = \\nabla_z \\log p(z) + \\text{Error}.
$$
Now consider the KL divergence between $q(y)$ and $p(y|z)$:
$$
\\mathcal{D}\_{\\text{KL}}(q(y)\\|p(y|z)) = \\int q(y) \\left[ \\log q(y) - \\log p(y|z) \\right] \\text{d}y.
$$
Taking the gradient with respect to $z$, we have
$$
\\nabla_z \\mathcal{D}\_{\\text{KL}}(q(y)\\|p(y|z)) = -\\int q(y) \\nabla_z \\log p(y|z) \\mathrm{d}y = -\\mathbb{E}\_{q(y)} \\left[ \\nabla_z \\log p(y|z) \\right],
$$
which is the negative of the error term above, proving the result. $\\Box$

Now note that early in sampling, the input $z$ is so noisy that the mutual information between $y$ and $z$ is negligible. In this case, $p(y|z)$ is very close to $p(y)$. (Think of it as a classifier with noisy input.) Accordingly, if we choose $q(y) = p(y)$, the expected error almost vanishes early in the sampling. Toward the end, the bias of the ICG estimate increases as $p(y)$ moves further from $p(y|z)$. However, since guidance is more important in early and mid stages of sampling [1, 2], we hypothesize that any increasing error in the unconditional score estimate is largely ignored by the model. We believe this is one of the core reasons why ICG gives similar results to CFG in practice.

[1] Balaji, Yogesh, Seungjun Nah, Xun Huang, Arash Vahdat, Jiaming Song, Qinsheng Zhang, Karsten Kreis et al. "ediff-i: Text-to-image diffusion models with an ensemble of expert denoisers." *arXiv preprint arXiv:2211.01324* (2022).

[2] Castillo, Angela, Jonas Kohler, Juan C. Pérez, Juan Pablo Pérez, Albert Pumarola, Bernard Ghanem, Pablo Arbeláez, and Ali Thabet. "Adaptive guidance: Training-free acceleration of conditional diffusion models." *arXiv preprint arXiv:2312.12487* (2023).

---

> ### Comment · Reviewer_8vmw · 2024-11-23
>
> I appreciate the analysis provided by the authors and their _thorough_ responses. I will raise my score after an answer to the following questions. I appreciate the authors patience
>
> While the mixture of Gaussians is a toy example, the disjoint support can easily arise solving inverse problems, or image-to-image generation, where $p(z_t | y) \approx 0$ for $y$ sampled independently, can the authors run the following experiment:
> 1. Take N conditions y where $y$ is an image vector, the authors can use their existing experimental setup for this as well.
> 2. For each t in $[T, \dots, 1, 0]$, compute the norm of the scores $s_\theta(z_t, y)$ and $s_\theta(z_t, \widehat{y})$.
> 3. report the average norm for each t for both scores.
> The reason I am asking for this last experiment is that I suspect that the norm of $s_\theta(z_t, \widehat{y})$ is near zero, which would imply ICG is equivalent to running scaled conditional guidance for some settings.
>
> Finally, can the authors provide some thoughts on the following questions:
> 1. If we are doing text to image generation, does ICG require us to keep a bank of text prompts for generation?
> 2. If we are doing image-to-image generation, does ICG require a bank of images for generation?
> 3. From their analysis, it seems that the authors would recommend sampling from a Normal distribution around $N(encoder(y), \sigma^2 I_d)$ or should we compute a global mean to be user for any conditioning variable $y$ as the paper suggested?
>
>
> As it stands, there are other approaches which propose alternatives to using a marginal score for guidance and have a clear algorithm and implementation. I think if the authors provide some analysis on different approaches for sampling $y$ and some analysis of the error, it would make the paper much stronger.

---

> > ### Comment · Reviewer_ryvA · 2024-11-23
> >
> > I thank the authors for performing this analysis, which I think provides much more insight into the properties ICG estimator. I will also increase my score, but will wait for the author's reply to Reviewer 8vmw's comment above, specifically the following comment:
> >
> > "As it stands, there are other approaches which propose alternatives to using a marginal score for guidance and have a clear algorithm and implementation. I think if the authors provide some analysis on different approaches for sampling and some analysis of the error, it would make the paper much stronger."
> >
> > I agree with this comment, and the author's answer will help gauge the significance of the paper's main contribution.

---

> > > ### Author Response · Authors · 2024-11-23
> > > **Official Response to Reviewers**
> > >
> > > We thank the reviewers for replying to our comment. Please find our answers below.
> > >
> > > ### **Question about the norm of the score functions**
> > > We conducted the requested experiment using our pose-to-image model across 16 different conditions with 75 sampling steps. We observed that the norm of the score function stays away from zero (in the range of $[50, 80]$ for this model) for both the conditional term $s(z_t, t, y)$ and the unconditional term $s(z_t, t, \hat{y})$. However, we noted that the norm of the update direction $s(z_t, t, y) - s(z_t, t, \hat{y})$ approaches zero relative to the norm of $s(z_t, t, y)$ as sampling progresses. This aligns with the observation that guidance plays a more significant role early in the sampling process, eventually converging toward the conditional prediction $s(z_t, t, y)$ in later stages.
> > >
> > > ### **Questions about choosing $\hat{y}$**
> > > 1. ICG does not require a bank of text prompts, although using one is a valid approach. Instead, we adopted a simpler strategy by constructing an embedding vector using the CLIP tokenizer. Pseudocode for this method is provided in Figure 11.
> > > 2. For image-to-Image models, we used random image-like vectors generated either with random integers or Gaussian noise. While ICG could potentially work with real data as the random-conditional input, we demonstrated that this simpler approach is effective in practice.
> > > 3. We believe both methods perform equally well in practice. For the paper, we computed the Gaussian noise statistics based on the current batch of condition embeddings $y$, i.e., sampling from $\mathcal{N}(\text{mean}(y_{\text{embed}}), \text{var}(y_{\text{embed}})\mathbf{I})$. However, this likely serves as an estimate of the global mean and variance of $y_{\text{embed}}$, and sampling directly from this global distribution would likely yield similar results.
> > >
> > > ### **Comment about alternative guidance approaches**
> > > We believe ICG offers one of the simplest methods for performing CFG that requires neither training nor fine-tuning of any model, and it needs minimal additional code for implementation. Additionally, we believe that ICG offers intuition into the inner workings of CFG. Besides this, we also introduced TSG, which enhances the quality of diffusion models while being more general than CFG and making no assumptions about the network architecture. We hope the reviewers will recognize the significance of our methods in this context, as both techniques are general and easy-to-implement, and they demonstrate competitive empirical performance / advantages across diverse models and datasets.
> > >
> > > If the reviewers have specific alternative methods in mind, we will carefully consider them for inclusion in the final version of the paper. In addition, we will expand the discussion on selecting $p(y)$ and incorporate a more detailed error analysis similar to what we presented in the rebuttal into the final manuscript.

---

> > > > ### Comment · Reviewer_8vmw · 2024-11-23
> > > >
> > > > If we don't have a batch of conditioning embeddings $y$, just a single sample $y$, then how are the mean and variance computed?

---

> ### Author Response · Authors · 2024-11-23
> **Official Response to Reviewer 8vmw**
>
> In this case, the `mean` and `std` functions do not include a batch dimension and instead focus on the scales along the embedding dimension (as each embedding is a $1 \times D$ vector). We did not find this to be an issue in practice.

---

> > ### Comment · Reviewer_8vmw · 2024-11-23
> >
> > In that case, is it true that the condition $\widehat{y} \perp z_t$ does not hold. This is not a bad thing, the method is simple and straightforward, I am just asking for some clarity.

---

> > > ### Author Response · Authors · 2024-11-23
> > > **Official Response to Reviewer 8vmw**
> > >
> > > We hypothesize that most of the independent behavior is maintained between $z_t$ and $y$, as $y$ is replaced with Gaussian noise, and the `mean` and `std` functions serve as approximations of the global `mean` and `std` across all classes. However, we acknowledge that some information about $y$ may still leak into $\widehat{y}$ in this process. Incorporating a preprocessing step, where the `mean` and `std` are computed over all classes beforehand, could further mitigate this dependency. Nevertheless, we observed that even this straightforward, on-the-fly computation suffices for ICG to perform effectively in practice.

---

> > > > ### Comment · Reviewer_8vmw · 2024-11-23
> > > >
> > > > One could write $\widehat{y} = y_{\text{embed}} + \varepsilon$, where $\varepsilon \sim N$.  Therefore, the independence claim cannot be made in this case.
> > > >
> > > > Can the authors make the plot of the norms of the marginal, conditional, and independent scores for image-to-image models for multiple conditions $y$ and $t$'s? These models are easily available on huggingface? Experiments like this could help shed light on what is going on, since in the single sample case, the ICG ~ conditional update with perturbed $y$.

---

> > > > > ### Author Response · Authors · 2024-11-24
> > > > > **Response to Reviewer 8vmw**
> > > > >
> > > > > The random condition is not computed by adding noise to the embedding vector $y_{\text{embed}}$. Instead, we draw $\hat{y} \sim \mathcal{N}(\mu, \sigma I)$, where $\mu = \text{mean}(y_{\text{embed}})$ and $\sigma = \text{std}(y_{\text{embed}})$ are scalars computed across the condition vector's dimensions.
> > > > >
> > > > > We conducted the same experiment using our own implementation of the pose-to-image model described in [1]. We used Gaussian noise for ICG, and 16 different conditions. The requested plot of norms versus time can be found at [this anonymous link](https://ibb.co/6nG19sf). Note that the norm of the ICG predictions closely aligns with the norm of the marginal predictions made by CFG.
> > > > >
> > > > > [1] Sadat, Seyedmorteza, et al. "CADS: Unleashing the Diversity of Diffusion Models through Condition-Annealed Sampling." *The Twelfth International Conference on Learning Representations*.

---

> ### Comment · Reviewer_8vmw · 2024-11-25
>
> Thank you for adding the requested plot.
>
> 1. how are the authors computing these scores? can they provide some details on how the curves were generated?
> 2. If the conditioning vector $y$ is not a scalar, but an image or text so a high-dimensional vector, so the method would use a Gaussian perturbation of $y_{\text{embed}}$ when there is a batch size of 1? I am surprised that is any different from conditional guidance.

---

> ### Author Response · Authors · 2024-11-25
> **Response to Reviewer 8vmw**
>
> We believe that we can address both of the reviewer’s questions by describing our procedure for the requested experiment. In the pose-to-image model, the pose is represented as a stick-figure image, which is a high-dimensional vector. This corresponds to the true condition $y$. We compute the (scalar) mean and standard deviation of the “pixel values” of $y$. To form $\\hat{y}$, we sample a Gaussian with the same mean and standard deviation to fill out a vector of the corresponding size. Each dimension of this vector is independently sampled from a Gaussian, so we are not perturbing $y$. This process is repeated at each sampling time step to produce a new $\hat{y}$ drawn from the same Gaussian. The plots we presented are the result of repeating this experiment over 16 pose conditions and averaging the norms of the scores calculated in each step.

---

> > ### Comment · Reviewer_8vmw · 2024-11-26
> >
> > The Gaussian sample you get is equal to $\widehat{y} = mean(y_{embed}) + \sigma(y_embed) \varepsilon$, and even if you average across dimensions, you do not get independent samples (atleast for 1 sample), which is different from the paper's claim about why their method works.
> >
> > I think the paper is good and can benefit from more analysis and better presentation. The questions raised here are not edge cases but cases that can arise in practice.
> >
> > The authors have not made it clear what is it that makes their method work, since $\widehat{y}$ is not independent of $z_t$ when there is only one $y$. I think this experiment should not be done while generating, rather can the authors sample real data point and then noise it at different timescales and make the same plot where they use only one condition $y$ to generate the $\widehat{y}$ sampling distribution, not the whole batch.

---

> ### Author Response · Authors · 2024-11-26
> **Response to Reviewer 8vmw (Part 1 of 2)**
>
> For illustration, let's say that a single condition $y$ is a $32 \\times 32$ "image." We reduce these 1024 numbers down to a scalar mean and a scalar standard deviation. (That is, these two statistics are computed across the 1024 dimensions of $y$ and not over a batch of conditions.) Let's say the mean and standard deviation are 0.24 and 2, respectively. We then sample 1024 i.i.d. samples from $\\mathcal{N}(0.24,2^2)$ to fill out the $32 \\times 32$ random condition $\\hat{y}$.
>
> Since in this example the two scalar parameters of the Gaussian used to sample $\\hat{y}$ are derived from the 1024-dimensional $y$, the reviewer is correct that $\\hat{y}$ is technically dependent on $y$  (and, by extension, $z\_t$), albeit very weakly. However, this procedure is done only to conveniently set the scale for sampling $\\hat{y}$ to keep it in the range of values the model saw during training. It is not actually a requirement of our method. One can also sample $\\hat{y}$ from a Gaussian with no reference to $y$ whatsoever as long as the range of values is reasonable.
>
> We are glad that the reviewer's impression of the paper remains positive, and we appreciate the level of interest shown to what makes it work so well. We assure the reviewer that these subtler points will be revisited and clarified in the final version of the paper.

---

> > ### Author Response · Authors · 2024-11-27
> > **Response to Reviewer 8vmw (Part 2 of 2)**
> >
> > In addition to our above response to the first part of the reviewer's comment, we conducted the experiment requested by the reviewer: selecting a single datum $x$ and its condition $y$, adding noise to $x$ to obtain $z_t$ for various time steps, and computing the norm of the prediction for conditional, unconditional, and independent score estimations. The results are available at [this anonymous link](https://ibb.co/V2RwFq0). Consistent with our previous observations, the independent estimation closely follows the norm of the unconditional estimation.

---

> > > ### Comment · Reviewer_8vmw · 2024-11-29
> > >
> > > thanks, i have raised my score.

---

### Public Comment · ~Seyedmorteza_Sadat1 · 2025-03-01
**Camera-ready update**

We would like to thank again the reviewers and the AC for supporting our submission. For the camera-ready version, we have reverted to our paper’s original title before submission, which also matches its arXiv version, as it more directly highlights our core contributions, particularly in rethinking classifier-free guidance without additional training requirements, while staying within the original scope of the paper.

---

### Meta-Review · Area_Chair_rS3E · 2024-12-17

**Metareview:**

The manuscript proposes a new approach for diffusion models that achieves the benefits of Classifier-Free Guidance (CFG) without the need to train both a conditional and unconditional model. Two novel methods are introduced: Independent Condition Guidance (ICG) and Time-Step Guidance (TSG). ICG eliminates the special training procedures required by CFG, thereby simplifying the training process for conditional diffusion models. TSG extends guidance techniques to unconditional diffusion models by leveraging time-step perturbation. Both methods are easy to implement, computationally efficient, and empirically validated to perform on par with or better than CFG across multiple tasks and benchmarks.

By addressing the training inefficiencies of CFG, ICG streamlines the implementation of conditional diffusion models without compromising performance, making it particularly relevant for large-scale and resource-constrained applications. TSG, on the other hand, extends guidance techniques to unconditional models. The experimental validation is thorough and well-executed, showing consistent performance across various models and tasks. The proposed techniques are straightforward to integrate into existing frameworks, ensuring their immediate applicability and impact for the broader community.

The reviewing panel unanimously recommends acceptance of this paper. The proposed methods address meaningful limitations of current techniques while maintaining computational efficiency and delivering strong empirical performance. The contributions are novel, broadly applicable, and of significant interest to the community.

**Additional Comments On Reviewer Discussion:**

During the rebuttal period, there was significant and constructive exchange between the reviewers and the authors. The authors provided clarifications on several statements made in the paper (eg. bounding the error on the unconditional score approximation, short computations of mixture models, etc..), addressing concerns and improving the overall clarity of their contributions. Additionally, the authors ran new experiments to further validate their methods (eg. comparisons with SAG and PAG using Stable Diffusion)

---

### Decision · Program_Chairs · 2025-01-22

Accept (Poster)